# TextTIGER: Text-based Intelligent Generation with Entity Prompt Refinement for Text-to-Image Generation

## Abstract

When generating images from prompts that include specific entities, the model must retain as much entity-specific knowledge as possible. However, there is a countless number of entities in the world, and new entities emerge; memorizing all of them completely is not realistic. To bridge this gap, our work proposes Text-based Intelligent Generation with Entity Prompt Refinement (TextTIGER). TextTIGER strengthens knowledge about entities that appear in the prompt by augmenting external information and then summarizes the expanded descriptions with large language models, preventing performance degradation that arises from excessively long inputs. To evaluate our method, we construct a new dataset consisting of captions, images, detailed descriptions, and lists of entities. Experiments with multiple image generation models show that TextTIGER improves image generation performance on widely used evaluation metrics compared with prompts that use captions alone. In addition, using Multimodal LLM (MLLM)-as-a-judge and human evaluation by multiple annotators, we demonstrate that our method consistently achieves higher scores, which underscores its effectiveness. These results show that strengthening entity-related descriptions, summarizing them, and refining prompts to an appropriate length leads to substantial improvements in image generation performance. We will release the created dataset and code upon acceptance.

**Suggested by Reviewer K3b8**
**Suggested by Reviewer BEK8**
**Suggested by Reviewer 9eNX**

## 1 Introduction

Text-to-Image generation is a task that generates images from a given text (Zhang et al., 2023b; Croitoru et al., 2023) with a wide range of applications, including concept image creation and diagram generation (Zhang et al., 2023a). To generate images from textual information, image generation models such as Stable Diffusion (Rombach et al., 2022) adopt an architecture that combines a text encoder with a diffusion model (Ho et al., 2020). These models require carefully designed prompts to reflect the intended image content (Jeon et al., 2025; Lyu et al., 2024; Zhan et al., 2024; Zhang et al., 2023b).

In this process, the model should retain as much entity-specific knowledge as possible to generate images that meet user expectations. Such entities include proper nouns in the prompt, such as names of rivers, castles, and mountains (Seyler et al., 2018; Yamada et al., 2020; 2018; 2017; Gabrilovich et al., 2007).[1]

However, even large-scale image generation models cannot fully retain such knowledge or continuously acquire the latest information, as it demands substantial costs, i.e., the need to keep crawling for up-to-date information and to continuously train billion-scale models. Understanding entities correctly plays a crucial role in aligning with user intent in tasks such as an advertisement image generation task (Du et al., 2024; Mita et al., 2024). For example, as shown in Figure 1, when the prompt "Giant's Castle" is given, an image

---

[1] In our study, we define an entity as a named entity at the proper expression level, referring to a specific instance such as "Golden Gate Bridge" rather than an abstract concept such as "bridge" (Seyler et al., 2018; Huang et al., 2026).

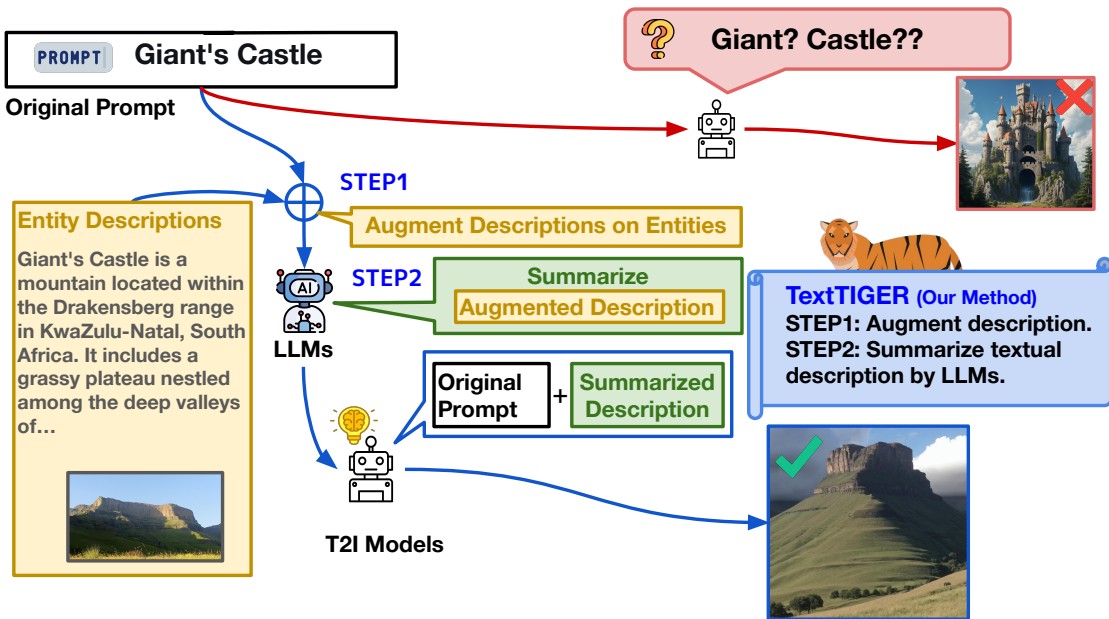

Figure 1: Overview of the proposed method. Our work (1) expands knowledge about entities and (2) summarizes the expanded descriptions to an appropriate length with LLMs, thereby improving the ability of image generation models to handle entities.

generation model may fail to interpret the entity correctly. Here, "Giant's Castle" refers to a mountain located in South Africa.[2] Moreover, simply appending externally retrieved information as a long context to the prompt does not enable effective and accurate processing, due to token length constraints such as a 512-token limit of text encoders (Tan et al., 2024; Zhang et al., 2024).

To address the limitation in entity understanding, we construct a new dataset consisting of image-caption pairs with annotated entity mentions that includes detailed descriptions for each entity, enabling systematic evaluation of how adding external knowledge about entities influences image generation quality. Based on this dataset, we propose a new method called Text-based Intelligent Generation with Entity Prompt Refinement (TEXTTIGER). Our method first retrieves entity-specific knowledge from external sources and expands the original prompt. For example, as shown in Figure 1, for the prompt "Giant's Castle," we retrieve additional context such as "Giant's Castle is a mountain located within the ...," which compensates for missing knowledge inside the model. Second, we leverage large language models (LLMs) (OpenAI et al., 2024; Singh et al., 2025; Yang et al., 2025; Grattafiori et al., 2024; Qwen et al., 2025) to summarize the retrieved descriptions concisely. This step preserves essential information while keeping the prompt within an appropriate token length. Finally, we generate images from these refined prompts, which mitigates both the model's knowledge limitations and the difficulty of processing long contexts.

Experiments with multiple image generation models together with LLMs on the constructed dataset show that our method substantially outperforms baseline approaches on widely used automatic evaluation metrics. While performance drops when we simply append descriptions, the performance improves when we summarize them, which supports the importance of concise descriptions of entities. Furthermore, evaluation results by Multimodal LLM (MLLM)-as-a-judge (Chen et al., 2024) and human evaluations by multiple annotators indicate that images generated from prompts with entity description summary contain more entity-related content and exhibit greater faithfulness.

---

[2]https://en.wikipedia.org/wiki/Giant%27s_Castle

## 2 Related Work

**Vision and Entity Knowledge** In the Vision and Language (V&L) field, challenges in understanding visual and textual information often reveal the limited generalization ability of V&L models in generating text from images for applications such as newspapers (Lu et al., 2018; Liu et al., 2021), e-commerce (Ma et al., 2022), fashion (Rostamzadeh et al., 2018), and artworks (Bai et al., 2021; Hayashi et al., 2024; Ozaki et al., 2025). Similarly, Kamigaito et al. (2023) show that the V&L model OFA (Wang et al., 2022) lacks sufficient entity knowledge in image generation tasks.

Several benchmarks also evaluate how well image generation models understand world knowledge (Chen et al., 2022; Zhao et al., 2025; Wu et al., 2025b; Niu et al., 2025). An extensive study by Chen et al. (2022) introduced "EntityDrawBench," a dataset covering 250 entities, and pointed out that image generation models lack knowledge about long-tail entities. Huang et al. (2026) introduced the "KITTEN" benchmark to evaluate knowledge-intensive generation and found that even the most advanced models often fail to generate entities with accurate visual details. In experiments across domains such as landmarks, plants, and animals, models including Stable Diffusion (Esser et al., 2024) and DALL-E 3 (Betker et al., 2023) produced images with substantial inaccuracies or missing critical features when asked to depict many real-world entities. These findings indicate that current diffusion models rely heavily on what they learn during training and lack robust factual grounding for many specific entities.

**Refinement of Image Generation Prompts** Researchers have shown that prompt engineering effectively improves image generation (Jeon et al., 2025; Lyu et al., 2024; Zhan et al., 2024). Zhan et al. (2024) refine prompts by training a dedicated text encoder with image representations to generate desired images. Other work proposes generating images by training models with reinforcement learning (Ghasemi et al., 2025; Schulman et al., 2017; Rafailov et al., 2023) so that they produce optimized prompts (Hao et al., 2023). Methods that refine prompts with external knowledge also exist. Jeong et al. (2025) point out that models lack cultural understanding and refine prompts with models equipped with external knowledge to produce more appropriate output images. Image generation approaches based on Retrieval-Augmented Generation (Chen et al., 2022) also attempt prompt refinement with retrievers, e.g., for abstract prompts (Lyu et al., 2024) and for multiple objects (Yuan et al., 2025).

However, although these approaches leverage external knowledge or iterative refinement to improve prompt quality, they do not explicitly focus on supplementing entity-level knowledge. In particular, prior methods do not retrieve and inject structured, entity-specific factual descriptions to compensate for missing world knowledge in text-to-image models, overlooking the limitation of input length by the text encoder as well. Instead, they often use external information, such as cultural alignment or safety refinement, without explicitly addressing whether the model has sufficient factual grounding about individual named entities.

## 3 Dataset Construction for Entity-Aware Image Generation

To evaluate whether providing rich descriptive information for named entities improves image generation quality, we construct a new dataset. The existing dataset PopVQA (Haklay et al., 2025) provides large-scale image-caption pairs but does not include explicit entity-level information. As a result, its usefulness is limited in settings where models must understand specific named entities and correctly align them with visual content. In real-world applications, prompts often include proper nouns and named entities that presuppose background knowledge. Without access to such knowledge, even advanced image generation models may hallucinate incorrect visual content, miss distinctive attributes, or confuse entities.

To address this issue, we extend the original PopVQA by adding background descriptions for all named entities that appear in each caption. We extend these descriptions through the Wikipedia API.[3] Specifically, the metadata of PopVQA contains hyperlinks to Wikipedia pages corresponding to entities mentioned in the captions. We systematically follow these URLs and extract the introductory abstract of each page. These introductory paragraphs typically provide concise and informative summaries, including the definition, classification, origin, and notable characteristics of the entity. Such abstracts serve as natural and

---

[3]https://www.mediawiki.org/wiki/API:Main_page

reliable sources of contextual knowledge, especially for uncommon, ambiguous, or culturally specific entities. For example, when given the caption "Liberty at sunset," the Wikipedia abstract provides supplementary information such as its location, height, appearance, and symbolic meaning. This knowledge often plays an important role in faithful image generation.

To ensure consistency and quality, we focus on the landmarks and paintings categories and retain only instances for which both the image and the linked Wikipedia page remain accessible at the time of dataset construction. Under these criteria, we extract 2,764 instances from the landmarks category and 2,245 instances from the paintings category, resulting in 5,009 valid instances in total. Each instance in our dataset consists of four elements: (1) the original image, (2) the corresponding caption, (3) the retrieved entity descriptions, and (4) the list of entities contained in those descriptions. The resulting new dataset enables controlled experimental analysis of how access to entity-specific background knowledge influences the behavior of text-to-image generation models. Details appear in Appendix C.8.

# 4 Proposed Method: TextTIGER

We propose a method that strengthens entity-specific knowledge by augmenting accurate descriptions of entities that appear in the prompt and then summarizing them to an appropriate length by LLMs, as shown in Figure 1. Our method effectively mitigates two major weaknesses of image generation models, i.e., (1) limited internal knowledge on entities, and (2) difficulty in handling long contexts. The proposed approach consists of the following 2 steps: STEP 1. augment entities with informative descriptions (§ 4.1), and STEP 2. summarize the descriptions with an LLM to the appropriate length (§ 4.2).

## 4.1 STEP 1: Augment Entities with Informative Descriptions

To enable image generation models to understand entities, we augment externally retrieved, information-rich descriptions to the entities that appear in the prompt. Specifically, we extract entities contained in the prompt and obtain corresponding descriptions to compensate for the model's limited internal knowledge.

## 4.2 STEP 2: Summarize the descriptions using LLMs

For the augmented entity-specific descriptions obtained in STEP 1, we use an LLM to generate summaries that preserve detailed entity information while keeping the length appropriate. Prior work (Juseon-Do et al., 2024) shows that explicitly specifying the input length and the desired number of output tokens helps LLMs manage length constraints effectively. From their motivation, we tokenize the augmented descriptions from STEP 1 using the tokenizer of the text encoder in the image generation model and explicitly provide the token count to the LLM, providing the detailed prompts in Appendix D.1. After this process, we concatenate the summarized entity-specific descriptions to the end of the original caption, forming a new prompt in the format "(caption + summarized description)" for image generation.

We refer to this as Text-based Intelligent Generation with Entity Prompt Refinement (TEXTTIGER).

# 5 Experimental Settings

## 5.1 Prompt Formats

To verify whether our proposed method, TEXTTIGER, properly improves entity-level image generation capability, we additionally compared 4 methods, as introduced in Table 1.

1. CAP-ONLY: This setting simply provides the caption that exists in the created dataset to image generation models, validating the baseline performance of image generation models.

2. AUG-ONLY: In this setting, we simply concatenate the entity descriptions to the caption, i.e., applying only the STEP 1 of our approach in § 4.1. We define this method to examine whether summarization is necessary.

Table 1: List of experimental settings and comparison methods in our study.

| Method | Prompt for Image Generation |
|---|---|
| **Cap-Only** | The caption in the dataset. |
| **Aug-Only** | The caption + Augmented knowledge from Wikipedia. |
| **RAG** | The caption + Augmented knowledge retrieved from the datastore. |
| **TextTIGER w/o Len** | The caption + Summarized description generated by LLMs. |
| **TextTIGER (Our proposed method)** | The caption + Summarized description generated by LLMs with the explicit token length. |

3. RAG: STEP 1 of our proposed method, i.e., § 4.1, supplements missing knowledge about entities by retrieving external information, which closely relates to the Retrieval-Augmented Generation (RAG) (Lewis et al., 2020) method. However, prior work shows that entities are more challenging to handle than general knowledge (Shachar et al., 2025). To evaluate performance under a RAG-based framework, we compare a method that uses BM25 (Lù, 2024) as the retriever, since preliminary experiments show that BM25 achieves the best retrieval performance as demonstrated in Appendix C.2. As the datastore, we use PopVQA data and additionally collect 109,598 Wikipedia articles related to landmarks and 132,573 articles related to paintings. Our method takes the caption as input and concatenates the description retrieved from the datastore to evaluate whether RAG improves performance in this setting.

4. TEXTTIGER W/O LEN: STEP 2 of our proposed method (§ 4.2) explicitly specifies the token length during summarization to generate prompts that suit image generation. To examine whether explicit token control improves prompt quality, we define a variant that performs summarization without specifying the token length. This comparison allows us to validate the effectiveness of STEP 2 and to examine whether we can apply prior findings (Juseon-Do et al., 2024) to prompt construction.

## 5.2 Summarization Models

To summarize to an appropriate length, we exploit LLMs with strong summarization capabilities, including Qwen2.5 (72B) (Qwen et al., 2025), Qwen3 (30B) (Yang et al., 2025), and Llama 3.3 (70B) (Grattafiori et al., 2024) for open models. As for proprietary models, we use GPT-4o (OpenAI et al., 2024) and GPT-5 (Singh et al., 2025) through API. Appendix C.1 provides detailed experimental settings.

## 5.3 Image Generation Models

To account for differences in text encoders, we use five image generation models. Specifically, we use Dreamlike 2.0 (defined as Dreamlike) (Art, 2023), which employs CLIP (Radford et al., 2021) as its text encoder, PixArt (Chen et al., 2023), which adopts T5 (Raffel et al., 2020), FLUX (Labs, 2024) and Stable Diffusion 3.5 (SD3.5) (Esser et al., 2024), both of which incorporate CLIP and T5, and Qwen-Image (defined as Qwen-Img) (Wu et al., 2025a), which uses Qwen (Bai et al., 2025) as its encoder. The maximum token lengths of the text encoders are 77 tokens for CLIP, 512 tokens for T5, and 4,096 tokens for Qwen-Image.

## 5.4 Evaluation Metrics

To measure how much entity-aware image generation improves, we adopt (1) automatic evaluation metrics, i.e., CLIPScore (Hessel et al., 2021), DINOScore (Oquab et al., 2024), and PickScore (Kirstain et al., 2023), and (2) Multimodal Large Language Models (MLLM)-as-a-judge (Chen et al., 2024) evaluated by Vision-Language Models (VLMs) (Qwen et al., 2025; Microsoft et al., 2025; Team et al., 2025).

**CLIPScore-T:** We utilize CLIPScore (Hessel et al., 2021), which computes the cosine similarity between the hidden states produced by the text encoder and the image encoder of CLIP (Radford et al., 2021) trained with contrastive learning on image-text pairs, to measure how faithfully an image reflects a given sentence. We define this metric as CLIPScore-T and use it to measure the similarity between the generated image and the entity used for generation.

Table 2: Results of experiments on the Landmarks category. We report CLIPScore-T and PickScore as evaluation metrics for text-to-image (Txt-Img) generation, and CLIPScore-I and DINOScore as evaluation metrics for image-to-image (Img-Img) generation. All values are reported as mean $\pm$ standard deviation. Statistical significance is evaluated using paired bootstrap resampling (10,000 samples) (Koehn, 2004). *, **, and *** denote $p < 0.05$, $p < 0.01$, and $p < 0.001$, respectively, compared to Cap-Only.

| T2I Model | Cap-Only | RAG | | TextTIGER w/o Len | | | | | TextTIGER (Proposed Method) | | | | |
|---|---|---|---|---|---|---|---|---|---|---|---|---|---|
| | | Aug-Only | BM25 | Qwen3 | Qwen2.5 | Llama3.3 | GPT-4o | GPT-5 | Qwen3 | Qwen2.5 | Llama3.3 | GPT-4o | GPT-5 |
| **CLIPScore-T** | | | | | | | | | | | | | |
| Dreamlike | $23.98_{+3.22}$ | $20.94_{+3.72}$ | $20.74_{+4.56}$ | $20.35_{+3.22}$ | $20.24_{+3.20}$ | $20.60_{+3.13}$ | $20.22_{+3.21}$ | $20.29_{+3.17}$ | $24.67^{***}_{+3.91}$ | $24.87^{***}_{+3.91}$ | $\mathbf{25.30^{***}_{+3.81}}$ | $24.85^{***}_{+3.84}$ | $24.83^{***}_{+3.86}$ |
| PixArt | $19.11_{+4.47}$ | $20.29^{***}_{+3.61}$ | $19.63^{***}_{+4.91}$ | $18.95_{+3.28}$ | $18.96_{+3.20}$ | $18.99_{+3.21}$ | $18.96_{+3.21}$ | $18.95_{+3.22}$ | $23.11^{***}_{+3.92}$ | $23.11^{***}_{+3.91}$ | $\mathbf{23.24^{***}_{+3.93}}$ | $23.18^{***}_{+3.85}$ | $23.14^{***}_{+3.90}$ |
| FLUX | $21.45_{+3.58}$ | $20.57^{***}_{+3.78}$ | $20.29^{***}_{+4.71}$ | $19.95_{+3.23}$ | $19.41_{+3.37}$ | $19.46_{+3.34}$ | $19.29_{+3.42}$ | $19.29_{+3.34}$ | $23.68^{***}_{+4.12}$ | $23.79^{***}_{+4.13}$ | $\mathbf{23.99^{***}_{+4.09}}$ | $23.69^{***}_{+4.15}$ | $23.67^{***}_{+4.12}$ |
| SD3.5 | $23.88_{+3.42}$ | $21.43_{+4.06}$ | $21.26_{+4.76}$ | $20.57_{+3.32}$ | $20.75_{+3.33}$ | $20.84_{+3.27}$ | $20.80_{+3.33}$ | $20.75_{+3.34}$ | $25.03^{***}_{+4.00}$ | $25.40^{***}_{+4.04}$ | $\mathbf{25.48^{***}_{+3.99}}$ | $25.37^{***}_{+3.96}$ | $25.33^{***}_{+3.95}$ |
| Qwen-Img | $22.42_{+4.23}$ | $17.47_{+5.61}$ | $17.71_{+5.53}$ | $20.09_{+3.44}$ | $19.92_{+3.41}$ | $19.89_{+3.41}$ | $19.95_{+3.46}$ | $19.91_{+3.42}$ | $24.34^{***}_{+4.20}$ | $24.34^{***}_{+4.28}$ | $24.23^{***}_{+4.33}$ | $\mathbf{24.44^{***}_{+4.12}}$ | $24.41^{***}_{+4.23}$ |
| **CLIPScore-I** | | | | | | | | | | | | | |
| Dreamlike | $68.37_{+10.90}$ | $66.48_{+10.12}$ | $65.09_{+10.57}$ | $64.36_{+7.95}$ | $64.56_{+7.81}$ | $64.35_{+8.15}$ | $64.36_{+7.99}$ | $64.39_{+7.95}$ | $78.67^{***}_{+9.80}$ | $\mathbf{79.07^{***}_{+9.73}}$ | $78.85^{***}_{+9.84}$ | $78.98^{***}_{+9.65}$ | $78.89^{***}_{+9.66}$ |
| PixArt | $59.68_{+11.53}$ | $68.51^{***}_{+9.24}$ | $66.43^{***}_{+10.28}$ | $62.48^{***}_{+8.11}$ | $62.73^{***}_{+7.91}$ | $62.34^{***}_{+8.05}$ | $62.59^{***}_{+7.91}$ | $62.59^{***}_{+7.89}$ | $75.86^{***}_{+9.71}$ | $76.25^{***}_{+9.91}$ | $75.44^{***}_{+9.90}$ | $\mathbf{76.70^{***}_{+9.64}}$ | $76.64^{***}_{+9.56}$ |
| FLUX | $66.52_{+11.61}$ | $69.14^{***}_{+10.48}$ | $67.46^{***}_{+11.19}$ | $64.28_{+9.02}$ | $64.11_{+8.59}$ | $63.84_{+8.61}$ | $63.74_{+8.65}$ | $63.77_{+8.58}$ | $77.93^{***}_{+10.50}$ | $\mathbf{78.43^{***}_{+10.58}}$ | $77.99^{***}_{+10.82}$ | $78.36^{***}_{+10.60}$ | $78.25^{***}_{+10.64}$ |
| SD3.5 | $68.99_{+11.51}$ | $67.45_{+11.03}$ | $66.04_{+11.37}$ | $64.76_{+8.20}$ | $65.41_{+8.18}$ | $65.42_{+8.23}$ | $65.30_{+8.24}$ | $65.34_{+8.18}$ | $79.33^{***}_{+10.00}$ | $\mathbf{79.94^{***}_{+10.05}}$ | $79.42^{***}_{+10.38}$ | $79.84^{***}_{+10.03}$ | $79.79^{***}_{+10.05}$ |
| Qwen-Img | $69.95_{+12.82}$ | $47.83_{+10.67}$ | $47.91_{+9.82}$ | $64.64_{+8.49}$ | $64.63_{+8.50}$ | $64.68_{+8.53}$ | $64.55_{+8.40}$ | $64.54_{+8.35}$ | $79.29^{***}_{+10.55}$ | $79.29^{***}_{+10.52}$ | $78.57^{***}_{+11.13}$ | $\mathbf{79.67^{***}_{+10.63}}$ | $79.56^{***}_{+10.72}$ |
| **DINOScore** | | | | | | | | | | | | | |
| Dreamlike | $29.94_{+24.86}$ | $27.05_{+23.20}$ | $23.89_{+22.51}$ | $33.25^{***}_{+21.14}$ | $33.42^{***}_{+21.19}$ | $34.03^{***}_{+20.90}$ | $33.55^{***}_{+21.26}$ | $33.50^{***}_{+21.17}$ | $45.24^{***}_{+28.06}$ | $\mathbf{45.56^{***}_{+28.25}}$ | $44.20^{***}_{+28.72}$ | $45.25^{***}_{+28.44}$ | $45.10^{***}_{+28.68}$ |
| PixArt | $19.03_{+20.40}$ | $34.84^{***}_{+22.64}$ | $29.15^{***}_{+23.26}$ | $34.86^{***}_{+19.11}$ | $35.12^{***}_{+19.07}$ | $34.68^{***}_{+19.12}$ | $35.10^{***}_{+19.35}$ | $35.04^{***}_{+19.25}$ | $45.94^{***}_{+25.56}$ | $\mathbf{46.31^{***}_{+25.77}}$ | $43.95^{***}_{+26.12}$ | $45.93^{***}_{+25.81}$ | $45.87^{***}_{+26.04}$ |
| FLUX | $30.02_{+25.47}$ | $37.44^{***}_{+25.09}$ | $32.63^{***}_{+25.32}$ | $39.92^{***}_{+21.71}$ | $37.74^{***}_{+20.87}$ | $37.57^{***}_{+20.79}$ | $37.70^{***}_{+21.11}$ | $37.65^{***}_{+20.94}$ | $48.97^{***}_{+27.97}$ | $\mathbf{50.33^{***}_{+28.13}}$ | $48.38^{***}_{+28.60}$ | $49.37^{***}_{+28.52}$ | $49.31^{***}_{+28.52}$ |
| SD3.5 | $36.49_{+25.59}$ | $35.25_{+24.54}$ | $30.68_{+24.59}$ | $39.47^{***}_{+20.92}$ | $40.46^{***}_{+20.87}$ | $40.58^{***}_{+20.78}$ | $40.32^{***}_{+20.95}$ | $40.45^{***}_{+21.01}$ | $52.59^{***}_{+27.98}$ | $\mathbf{53.64^{***}_{+28.15}}$ | $51.94^{***}_{+28.87}$ | $53.29^{***}_{+28.53}$ | $53.18^{***}_{+28.58}$ |
| Qwen-Img | $40.42_{+29.50}$ | $20.24_{+22.50}$ | $18.20_{+20.79}$ | $41.46^{***}_{+21.88}$ | $40.93^{***}_{+22.03}$ | $41.37^{***}_{+21.96}$ | $41.42^{***}_{+22.48}$ | $41.44^{***}_{+22.30}$ | $54.72^{***}_{+30.32}$ | $54.03^{***}_{+29.89}$ | $52.20^{***}_{+30.95}$ | $\mathbf{54.74^{***}_{+30.74}}$ | $54.43^{***}_{+30.75}$ |
| **PickScore** | | | | | | | | | | | | | |
| Dreamlike | $20.56_{+0.78}$ | $19.09_{+0.85}$ | $19.93_{+0.90}$ | $18.52_{+0.72}$ | $18.56_{+0.71}$ | $18.52_{+0.72}$ | $18.58_{+0.70}$ | $18.59_{+0.70}$ | $24.76^{***}_{+0.95}$ | $24.78^{***}_{+0.94}$ | $24.74^{***}_{+0.95}$ | $24.82^{***}_{+0.95}$ | $\mathbf{24.82^{***}_{+0.94}}$ |
| PixArt | $20.49_{+0.99}$ | $19.73_{+0.82}$ | $20.43_{+1.01}$ | $18.76_{+0.73}$ | $18.78_{+0.73}$ | $18.75_{+0.72}$ | $18.76_{+0.73}$ | $18.77_{+0.72}$ | $25.00^{***}_{+0.97}$ | $\mathbf{25.02^{***}_{+0.97}}$ | $25.02^{***}_{+0.99}$ | $24.95^{***}_{+0.98}$ | $24.94^{***}_{+0.98}$ |
| FLUX | $20.84_{+0.87}$ | $19.84_{+0.96}$ | $20.60_{+1.08}$ | $19.01_{+0.75}$ | $18.91_{+0.79}$ | $18.94_{+0.79}$ | $18.94_{+0.78}$ | $18.93_{+0.78}$ | $\mathbf{25.18^{***}_{+1.06}}$ | $25.16^{***}_{+1.06}$ | $25.13^{***}_{+1.06}$ | $25.17^{***}_{+1.05}$ | $25.17^{***}_{+1.05}$ |
| SD3.5 | $20.53_{+0.73}$ | $19.14_{+0.87}$ | $19.96_{+0.89}$ | $18.55_{+0.73}$ | $18.59_{+0.72}$ | $18.59_{+0.71}$ | $18.60_{+0.72}$ | $18.60_{+0.72}$ | $24.70^{***}_{+0.98}$ | $\mathbf{24.75^{***}_{+0.95}}$ | $24.66^{***}_{+0.98}$ | $24.73^{***}_{+0.95}$ | $24.73^{***}_{+0.94}$ |
| Qwen-Img | $20.61_{+0.93}$ | $17.89_{+0.81}$ | $18.79_{+0.74}$ | $18.80_{+0.81}$ | $18.49_{+0.72}$ | $18.68_{+0.82}$ | $18.51_{+0.71}$ | $18.51_{+0.70}$ | $\mathbf{24.93^{***}_{+1.12}}$ | $24.62^{***}_{+0.98}$ | $24.59^{***}_{+1.07}$ | $24.62^{***}_{+0.99}$ | $24.62^{***}_{+0.99}$ |

**CLIPScore-I:** Likewise, we compute the similarity between two different images using the hidden states produced by the image encoder of CLIP (Radford et al., 2021). Our work defines this metric as CLIPScore-I and uses it to measure the similarity between the reference image and the generated image.

**DINOScore:** We also compute image-image similarity using the DINO image encoder (Oquab et al., 2024), which they claim to outperform CLIP. As in CLIPScore-I, we measure the similarity between the reference image and the generated image. We define this similarity metric based on DINO as DINOScore.

**PickScore:** We adopt PickScore (Kirstain et al., 2023), which trains on human preference data and estimates the probability that an image aligns with a given textual instruction in a way humans prefer. PickScore builds on CLIP and, like CLIPScore-T, evaluates the similarity between text and image.

In our work, we use these four automatic metrics to evaluate whether image generation models improve their ability to generate entities accurately.

**Human Evaluation** We conduct human evaluation using Amazon Mechanical Turk (MTurk) (Crowston, 2012) with multiple annotators. Specifically, annotators are shown a reference image along with images generated by each method, and they evaluate the pairwise results on a 1—5 scale. Appendices C.6 and D.3 provide the detailed instruction and cost information.

**MLLM-as-a-judge:** Huang et al. (2026) proposed "KITTEN," an evaluation framework based on Multi-modal LLM (MLLM)-as-a-judge (Chen et al., 2024) for evaluating entity-level fidelity in generated images, and reported a high correlation with human evaluation, claiming that it serves as an effective substitute for manual assessment. In our study, we also adopt KITTEN to evaluate whether our proposed method, TEXT-TIGER, improves entity-aware image generation, and the detailed prompts are provided in Appendix D.2. Specifically, following KITTEN, we conduct an MLLM-as-a-judge evaluation from two perspectives: **Entity Alignment** and **Text Alignment**. **Entity Alignment** measures how accurately the generated image reflects the target entity, given a reference image that contains the entity, and VLMs rate this aspect on a 1–5 scale. **Text Alignment** measures how faithfully the generated image follows the input text prompt, and VLMs also rate this aspect on a 1–5 scale. As VLMs that serve as evaluators and support multiple image

Table 3: Experimental results on the Paintings category. The interpretation is the same as in Table 2.

| T2I Model | Cap-Only | RAG | | TextTIGER w/o Len | | | | | TextTIGER (Proposed Method) | | | | |
|---|---|---|---|---|---|---|---|---|---|---|---|---|---|
| | | Aug-Only | BM25 | Qwen3 | Qwen2.5 | Llama3.3 | GPT-4o | GPT-5 | Qwen3 | Qwen2.5 | Llama3.3 | GPT-4o | GPT-5 |
| **CLIPScore-T** | | | | | | | | | | | | | |
| Dreamlike | $23.83_{\pm3.90}$ | $19.83_{\pm4.80}$ | $19.71_{\pm4.92}$ | $21.96_{\pm2.84}$ | $21.83_{\pm2.37}$ | $21.53_{\pm2.43}$ | $21.34_{\pm2.48}$ | $21.57_{\pm2.36}$ | $24.41^{***}_{\pm4.75}$ | $24.61^{***}_{\pm4.52}$ | $\mathbf{24.71^{***}_{\pm4.76}}$ | $24.49_{\pm4.63}$ | $24.44_{\pm4.65}$ |
| PixArt | $21.46_{\pm4.50}$ | $20.56_{\pm4.24}$ | $19.93_{\pm4.77}$ | $21.38_{\pm2.35}$ | $21.20_{\pm2.48}$ | $21.11_{\pm2.83}$ | $20.75_{\pm3.26}$ | $20.76_{\pm2.68}$ | $23.81^{***}_{\pm4.48}$ | $23.87^{***}_{\pm4.48}$ | $\mathbf{23.93^{***}_{\pm4.52}}$ | $23.66_{\pm4.50}$ | $23.70_{\pm4.52}$ |
| FLUX | $21.60_{\pm3.65}$ | $20.14_{\pm4.17}$ | $19.91_{\pm4.60}$ | $21.60_{\pm2.77}$ | $21.28_{\pm2.56}$ | $21.52_{\pm2.87}$ | $21.00_{\pm2.72}$ | $21.08_{\pm3.15}$ | $23.29^{***}_{\pm4.31}$ | $\mathbf{23.56^{***}_{\pm4.23}}$ | $23.55^{***}_{\pm4.28}$ | $23.20_{\pm4.32}$ | $23.20_{\pm4.31}$ |
| SD3.5 | $23.57_{\pm3.72}$ | $20.88_{\pm4.68}$ | $20.45_{\pm4.93}$ | $21.95_{\pm2.78}$ | $21.64_{\pm2.73}$ | $22.03_{\pm2.13}$ | $21.98_{\pm2.13}$ | $22.30_{\pm2.30}$ | $24.26^{***}_{\pm4.40}$ | $24.44^{***}_{\pm4.46}$ | $\mathbf{24.60^{***}_{\pm4.42}}$ | $24.13_{\pm4.36}$ | $24.13_{\pm4.39}$ |
| Qwen-Img | $21.38_{\pm4.43}$ | $18.27_{\pm5.04}$ | $18.43_{\pm5.04}$ | $21.65_{\pm2.94}$ | $21.97_{\pm2.05}$ | $21.28_{\pm3.30}$ | $21.32_{\pm3.00}$ | $21.89_{\pm2.72}$ | $\mathbf{24.21^{***}_{\pm4.63}}$ | $23.59^{***}_{\pm4.54}$ | $23.64^{***}_{\pm4.68}$ | $23.32_{\pm4.60}$ | $23.31_{\pm4.57}$ |
| **CLIPScore-I** | | | | | | | | | | | | | |
| Dreamlike | $55.86_{\pm13.50}$ | $58.17^{***}_{\pm12.78}$ | $57.85^{***}_{\pm11.68}$ | $56.19_{\pm7.56}$ | $56.65_{\pm8.02}$ | $56.31_{\pm8.03}$ | $55.95_{\pm8.48}$ | $55.93_{\pm8.91}$ | $\mathbf{67.42^{***}_{\pm13.12}}$ | $66.48^{***}_{\pm12.83}$ | $66.97^{***}_{\pm13.26}$ | $65.77_{\pm12.30}$ | $65.74_{\pm12.43}$ |
| PixArt | $56.31_{\pm9.98}$ | $59.75^{***}_{\pm11.77}$ | $58.45^{***}_{\pm11.53}$ | $57.23_{\pm8.84}$ | $56.30_{\pm8.80}$ | $56.58_{\pm8.87}$ | $56.11_{\pm8.62}$ | $55.57_{\pm8.88}$ | $\mathbf{66.71^{***}_{\pm12.04}}$ | $66.06^{***}_{\pm11.55}$ | $65.90^{***}_{\pm11.98}$ | $65.38_{\pm11.33}$ | $65.46_{\pm11.46}$ |
| FLUX | $53.71_{\pm11.05}$ | $57.52^{***}_{\pm11.85}$ | $57.14^{***}_{\pm11.59}$ | $56.77_{\pm8.79}$ | $56.94_{\pm8.88}$ | $56.91_{\pm8.95}$ | $56.42_{\pm9.09}$ | $57.00_{\pm8.68}$ | $\mathbf{63.91^{***}_{\pm11.35}}$ | $62.96^{***}_{\pm11.49}$ | $62.84^{***}_{\pm11.58}$ | $62.01_{\pm11.06}$ | $61.95_{\pm11.06}$ |
| SD3.5 | $56.43_{\pm11.51}$ | $57.53^{***}_{\pm13.55}$ | $57.79^{***}_{\pm12.53}$ | $58.19_{\pm8.62}$ | $57.51_{\pm8.50}$ | $57.68_{\pm8.82}$ | $58.19_{\pm8.03}$ | $57.42_{\pm8.70}$ | $\mathbf{66.25^{***}_{\pm12.25}}$ | $65.56^{***}_{\pm12.15}$ | $65.77^{***}_{\pm12.44}$ | $64.52_{\pm11.77}$ | $64.47_{\pm11.86}$ |
| Qwen-Img | $56.20_{\pm12.21}$ | $49.76_{\pm14.84}$ | $51.66_{\pm13.68}$ | $56.16_{\pm8.21}$ | $56.09_{\pm7.00}$ | $56.73_{\pm7.94}$ | $55.21_{\pm7.96}$ | $56.05_{\pm6.98}$ | $\mathbf{68.86^{***}_{\pm13.58}}$ | $67.62^{***}_{\pm13.23}$ | $66.91^{***}_{\pm13.52}$ | $67.12_{\pm13.05}$ | $67.15_{\pm12.99}$ |
| **DINOScore** | | | | | | | | | | | | | |
| Dreamlike | $30.67_{\pm26.19}$ | $27.45_{\pm27.09}$ | $26.64_{\pm26.09}$ | $32.05_{\pm24.17}$ | $34.95_{\pm26.84}$ | $36.70_{\pm26.79}$ | $34.04_{\pm27.57}$ | $32.96_{\pm27.71}$ | $\mathbf{44.32^{***}_{\pm32.06}}$ | $42.48^{***}_{\pm31.16}$ | $43.62^{***}_{\pm31.95}$ | $41.01_{\pm30.87}$ | $40.73_{\pm30.82}$ |
| PixArt | $30.62_{\pm21.72}$ | $39.98^{***}_{\pm25.27}$ | $37.74^{***}_{\pm25.58}$ | $35.49_{\pm26.28}$ | $34.76_{\pm24.40}$ | $37.39_{\pm24.69}$ | $34.54_{\pm25.46}$ | $35.31_{\pm25.66}$ | $\mathbf{44.04^{***}_{\pm30.14}}$ | $42.79^{***}_{\pm28.29}$ | $43.62^{***}_{\pm29.12}$ | $41.33_{\pm28.10}$ | $41.30_{\pm28.06}$ |
| FLUX | $24.12_{\pm22.08}$ | $32.08^{***}_{\pm24.56}$ | $30.60^{***}_{\pm23.99}$ | $36.91_{\pm28.23}$ | $\mathbf{38.22^{***}_{\pm27.38}}$ | $36.27_{\pm27.27}$ | $36.39_{\pm28.78}$ | $36.41_{\pm28.75}$ | $33.84^{***}_{\pm25.36}$ | $33.46^{***}_{\pm24.53}$ | $34.05^{***}_{\pm25.85}$ | $31.52_{\pm24.28}$ | $31.34_{\pm24.07}$ |
| SD3.5 | $30.47_{\pm26.12}$ | $32.52^{***}_{\pm28.52}$ | $32.22^{***}_{\pm27.17}$ | $38.88_{\pm28.54}$ | $39.30_{\pm27.40}$ | $39.07_{\pm27.52}$ | $37.48_{\pm28.89}$ | $38.32_{\pm29.13}$ | $39.37^{***}_{\pm28.94}$ | $38.31^{***}_{\pm28.02}$ | $\mathbf{40.36^{***}_{\pm29.75}}$ | $35.85_{\pm27.59}$ | $35.84_{\pm27.56}$ |
| Qwen-Img | $33.03_{\pm25.33}$ | $24.97_{\pm23.51}$ | $26.04_{\pm22.98}$ | $39.76_{\pm29.36}$ | $39.41_{\pm26.10}$ | $40.56_{\pm28.42}$ | $38.34_{\pm28.42}$ | $39.34_{\pm27.53}$ | $\mathbf{46.23^{***}_{\pm31.75}}$ | $46.17^{***}_{\pm30.37}$ | $45.63^{***}_{\pm30.66}$ | $45.07_{\pm30.26}$ | $45.34_{\pm30.37}$ |
| **PickScore** | | | | | | | | | | | | | |
| Dreamlike | $20.20_{\pm0.99}$ | $18.54_{\pm1.09}$ | $19.33_{\pm1.07}$ | $18.95_{\pm0.66}$ | $19.04_{\pm0.64}$ | $19.03_{\pm0.59}$ | $19.03_{\pm0.62}$ | $19.04_{\pm0.65}$ | $24.34^{***}_{\pm1.17}$ | $24.33^{***}_{\pm1.17}$ | $24.28^{***}_{\pm1.18}$ | $\mathbf{24.37^{***}_{\pm1.19}}$ | $24.36_{\pm1.19}$ |
| PixArt | $20.68_{\pm1.30}$ | $19.28_{\pm1.11}$ | $19.91_{\pm1.19}$ | $19.35_{\pm0.61}$ | $19.33_{\pm0.61}$ | $19.38_{\pm0.74}$ | $19.28_{\pm0.66}$ | $19.25_{\pm0.71}$ | $24.73^{***}_{\pm1.31}$ | $\mathbf{24.83^{***}_{\pm1.33}}$ | $24.77^{***}_{\pm1.35}$ | $24.81_{\pm1.32}$ | $24.81_{\pm1.32}$ |
| FLUX | $20.86_{\pm1.14}$ | $19.54_{\pm1.16}$ | $20.17_{\pm1.30}$ | $19.51_{\pm0.76}$ | $19.66_{\pm0.61}$ | $19.60_{\pm0.78}$ | $19.55_{\pm0.66}$ | $19.56_{\pm0.62}$ | $24.85^{***}_{\pm1.31}$ | $24.91^{***}_{\pm1.33}$ | $24.89^{***}_{\pm1.33}$ | $\mathbf{24.92^{***}_{\pm1.35}}$ | $24.92_{\pm1.34}$ |
| SD3.5 | $20.36_{\pm0.97}$ | $18.86_{\pm1.13}$ | $19.65_{\pm1.17}$ | $19.20_{\pm0.69}$ | $19.21_{\pm0.60}$ | $19.03_{\pm0.76}$ | $19.12_{\pm0.63}$ | $19.16_{\pm0.64}$ | $24.48^{***}_{\pm1.23}$ | $\mathbf{24.51^{***}_{\pm1.24}}$ | $24.45^{***}_{\pm1.22}$ | $24.51_{\pm1.24}$ | $24.48_{\pm1.24}$ |
| Qwen-Img | $20.28_{\pm1.31}$ | $18.11_{\pm1.36}$ | $18.97_{\pm1.30}$ | $19.40_{\pm0.53}$ | $19.25_{\pm0.84}$ | $19.38_{\pm1.04}$ | $18.74_{\pm0.85}$ | $19.09_{\pm0.72}$ | $\mathbf{24.58^{***}_{\pm1.38}}$ | $23.95^{***}_{\pm1.28}$ | $24.00^{***}_{\pm1.35}$ | $23.89_{\pm1.22}$ | $23.90_{\pm1.21}$ |

inputs, we use 3 models: Qwen 2.5 (Qwen et al., 2025), Phi 4 (Microsoft et al., 2025), and Gemma 3 (Team et al., 2025). In our work, to address the issue that VLMs do not sufficiently understand entities (Hayashi et al., 2024; Ozaki et al., 2025; Alonso et al., 2025), we insert a brief description of each entity into the evaluation prompt when applying MLLM-as-a-judge. Appendix C.1 provides the details of each model, and Appendix D.2 shows the specific prompts we use.

To ensure the robustness of our evaluation under the stochastic nature of text-to-image generation, we apply bootstrap sampling to both the MLLM-as-a-judge scores and the automatic evaluation metrics. Following prior work (Kamigaito et al., 2023), we conduct paired bootstrap resampling (Koehn, 2004) with 10,000 samples to estimate the mean, standard deviation, and statistical significance.

## 6 Results and Discussions

Tables 2 and 3 present the results of the automatic evaluation metrics as described in § 5.4. In general, focusing on the Landmarks category in Table 2, TextTIGER achieves the best performance across the metrics, as indicated by the **bold** scores in the table, showing that augmenting entity-specific content and summarizing it to an appropriate length effectively improves entity-aware image generation.

In contrast, the Aug-Only setting shows little improvement over Cap-Only, indicating that simply appending additional knowledge does not lead to better performance. Moreover, when we compare TextTIGER w/o Len with the TextTIGER, TextTIGER consistently performs better, supporting the importance of explicitly controlling the summary length to construct prompts that better suit image generation models. We observe the same trend in the Paintings category shown in Table 3, and these consistent improvements across both domains demonstrate the generalization ability of our method.

### 6.1 Does TextTIGER Perform Effectively?

We showed that TextTIGER shows clear improvements when we compare the baseline. Taking a closer look at the result, for CLIPScore-T, PixArt improves from 21.459 to 23.930 with summaries generated by Llama 3.3, and Qwen-Img improves from 21.377 to 24.205 with summaries generated by Qwen 3. For DINOScore, Qwen-Img improves from 30.465 under Cap-Only to 46.228 when we use the optimized prompt summarized by Qwen 3.

Table 4: Token length statistics (mean$_{\pm\text{std}}$). **Bold** indicates the results of TEXTTIGER.

| Model | Landmarks | | | Paintings | | |
|---|---|---|---|---|---|---|
| | Aug-Only | w/o Len | TextTIGER | Aug-Only | w/o Len | TextTIGER |
| Qwen3 | $733.2_{\pm408.1}$ | $190.2_{\pm18.1}$ | $\mathbf{75.4}_{\pm15.1}$ | $467.5_{\pm384.0}$ | $186.3_{\pm24.2}$ | $\mathbf{74.8}_{\pm20.1}$ |
| Llama 3.3 | $707.5_{\pm393.9}$ | $96.9_{\pm22.8}$ | $\mathbf{33.9}_{\pm9.0}$ | $454.9_{\pm373.7}$ | $88.8_{\pm27.9}$ | $\mathbf{31.1}_{\pm9.1}$ |
| Qwen 2.5 | $733.2_{\pm408.1}$ | $103.9_{\pm20.4}$ | $\mathbf{40.1}_{\pm7.5}$ | $467.5_{\pm384.0}$ | $100.5_{\pm23.4}$ | $\mathbf{39.2}_{\pm8.9}$ |
| GPT-4o/5 | $693.6_{\pm387.3}$ | $105.2_{\pm14.1}$ | $\mathbf{45.9}_{\pm5.5}$ | $446.0_{\pm368.3}$ | $112.4_{\pm20.2}$ | $\mathbf{45.4}_{\pm5.7}$ |

Next, we examine whether simply expanding entity knowledge enables image generation models to understand entities. When we compare RAG setting with TEXTTIGER, TEXTTIGER consistently achieves better overall performance. For example, in CLIPScore-I, FLUX achieves 57.525 under the RAG-style expansion, whereas LLM-based summarization raises the score up to 68.857, yielding an improvement of more than 10 points. Across other automatic evaluation metrics, TEXTTIGER also outperforms CAP-ONLY, which demonstrates that TEXTTIGER effectively enhances entity-aware image generation.

We also analyze whether explicit length control during summarization is necessary. When we compare TEXTTIGER W/O LEN with TEXTTIGER, the TEXTTIGER consistently achieves better performance. Moreover, TEXTTIGER W/O LEN performs at a level comparable to CAP-ONLY, which highlights that instructing LLMs to summarize without explicit token-length control can even degrade performance. In contrast, TEXTTIGER improves performance, proving that image generation models remain highly sensitive to token length, yet can substantially benefit when the refined prompt fits within an appropriate input length.

Table 4 reports the measured prompt token lengths for three knowledge expansion methods: AUG-ONLY, TEXTTIGER W/O LEN, and TEXTTIGER. We compute the token counts using the tokenizer of each summarization model. For the proprietary GPT-4o/5 models, we use the `tiktoken` library (`https://github.com/openai/tiktoken`). The results in the table further confirm that explicitly controlling the token length during summarization, as proposed by Juseon-Do et al. (2024), improves the performance.

## 6.2 Do Encoders Understand Entities?

Our work showed that supplementing entity information with external knowledge improves image generation performance. We now ask how well the text encoders of image generation models internally represent entities themselves. To answer this question, we adopt a knowledge probing framework (Kalo & Fichtel, 2022; Wiland et al., 2024; Petroni et al., 2019; Youssef et al., 2023; Jinno et al., 2026). Our work constructs an entity retrieval task using the hidden states produced by each text encoder and evaluates performance with Hits@$k$ ($k \in \{1, 5, 10, 100\}$ in our work). Specifically, we input a description of each entity and use its embedding representation to retrieve the corresponding gold entity from a datastore. For the datastore, we take the 5,009 descriptions constructed in § 3 and rewrite them into ambiguous expressions using GPT-5 so that they do not directly contain the gold entity names, guaranteeing semantic entity understanding evaluation.

Figure 2 presents the results. For the Landmarks category (top row), the default encoders of Dreamlike, PixArt, FLUX, and Qwen-Img achieve almost zero Hits@1 and Hits@5, and even Hits@100 remains extremely low. When we use CLIP-G and CLIP-L text encoders in SD3.5, performance improves to some extent. However, even then, Hits@100 stays around 0.7, and accuracy at smaller $k$ remains low. This observation is the same trend in the Paintings category (bottom row). The text encoders of image generation models fail to reliably identify the correct entity from ambiguous descriptions. In contrast, the sentence embedding models introduced for comparison, SimCSE (Gao et al., 2021) and RoBERTa (Liu et al., 2019), achieve much higher performance. In Landmarks, SimCSE exceeds 0.8 in Hits@100 and clearly outperforms image generation encoders in Hits@10 and Hits@5 as well. We observe the same pattern in Paintings, where SimCSE consistently shows strong retrieval performance, highlighting the difference between text encoders trained for NLP tasks and those used in image generation models. These findings suggest that although text encoders in image generation models suffice for image conditioning, they struggle with knowledge-intensive reasoning that requires uniquely identifying entities from ambiguous descriptions. In other words, their

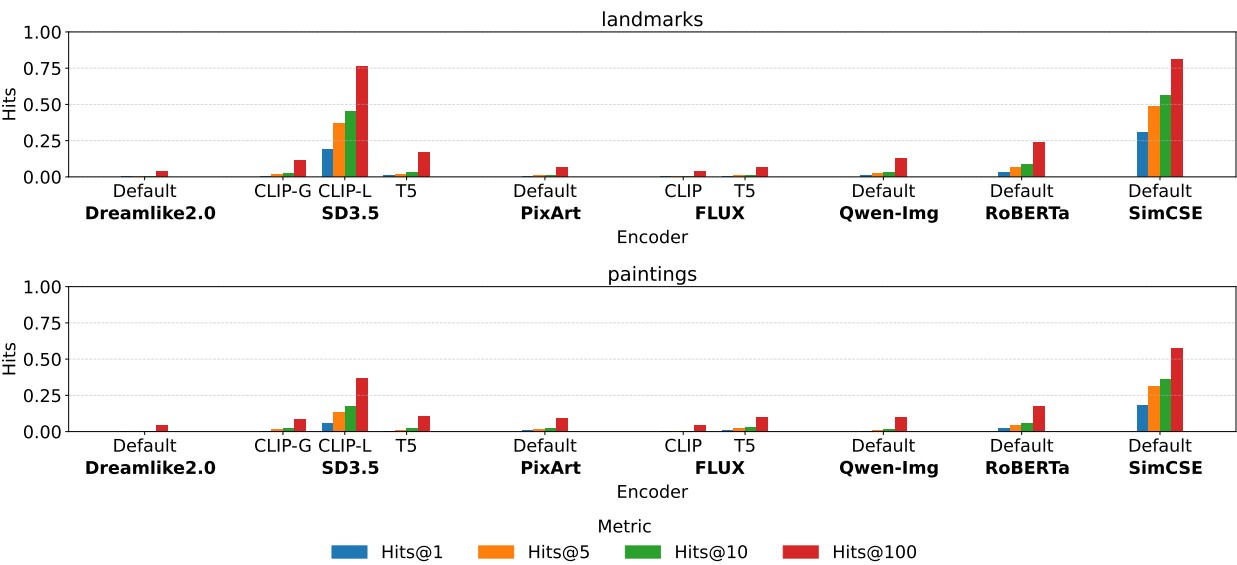

Figure 2: Visualization of how well the text encoders of image generation models understand entities. We examine whether each model can retrieve the correct description when given an entity as input.

Table 5: Human evaluation results on 100 randomly sampled instances for each category. **Bold values** indicate the best-performing result in each row. Humans (Avg.) indicates the average of five people.

| Category | Model | | | | | | | Humans Avg. |
|---|---|---|---|---|---|---|---|---|
| | **Dreamlike** | **PixArt** | **FLUX** | **SD3.5** | **Qwen-Img** | **SimCSE** | **RoBERTa** | |
| Landmark | 0.000 | 0.061 | 0.011 | 0.060 | 0.042 | 0.225 | 0.223 | **0.725** |
| Paintings | 0.000 | 0.008 | 0.052 | 0.067 | 0.083 | 0.295 | 0.238 | **0.723** |

internal entity knowledge remains limited, supporting our motivation that supplementing entity knowledge with external information effectively addresses this limitation.

We also conducted a human evaluation through MTurk to evaluate the quality of the retrieved entities by text encoders. We randomly sampled 100 descriptions from each category, i.e., 200 instances in total, and hired up to five annotators. For each ambiguous abstract, we presented the top-4 entities retrieved by SimCSE as candidates since SimCSE provides the best performance as shown in Figure 2, along with a "None of the above" option. Appendix C.6 provides the detailed annotation costs and guidelines.

The results in Table 5 show that human annotators can reliably identify the correct entity from the ambiguous abstract with 5 options, indicating that the retrieval task itself is feasible for humans and that the ambiguous descriptions preserve sufficient semantic information for entity disambiguation. This finding further supports our interpretation that the low retrieval accuracy observed for image generation encoders mainly stems from limited entity understanding capabilities rather than flaws in the probing setup itself.

### 6.3 Do Descriptions Support Unknown Entities?

To verify whether our augmented approach, as introduced in § 4.1, truly benefits "unknown" entities, we split entities into two groups based on the retrieval results in § 6.2 under the CAP-ONLY setting, i.e., "known" (Hits=1) and "unknown" (Hits=0). In Hits@1 of § 6.2, an entity is categorized as "known" when the encoder successfully retrieves the corresponding gold entity with the highest probability given an ambiguous abstract as input; otherwise, it is categorized as "unknown." Similarly, in Hits@$k$, an entity is categorized as "known" when the gold entity is included within the top-$k$ retrieved results. We then computed the difference in DINOScore as introduced in § 5.4 between CAP-ONLY and TEXTTIGER as $\Delta$DINO and

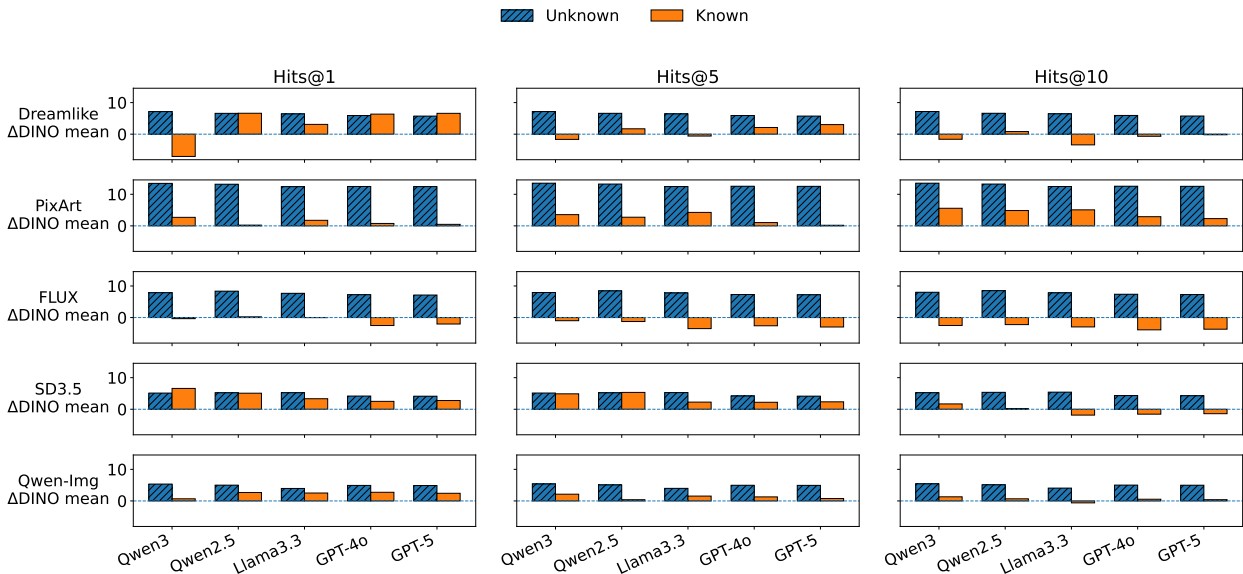

Figure 3: Visualization of whether our proposed method, TEXTTIGER, effectively handles unknown entities. The *y*-axis shows the DINOScore difference between CAP-ONLY and TEXTTIGER. The blue bins represent the subset of previously unknown entities, and the orange bins represent the subset of known entities.

analyzed how the effect varies depending on whether the entity was known or unknown. Figure 3 presents the results. [4] The most notable observation is that for Hits=0, namely cases where the text encoder failed to correctly identify the entity (blue bars with diagonal line), $\Delta$DINO consistently shows positive values across many models. In particular, PixArt and FLUX exhibit large positive improvements for Hits@1, Hits@5, and Hits@10, indicating that adding external descriptions markedly improves visual alignment when the model lacks sufficient internal entity knowledge. This finding demonstrates that even when the model does not internally encode adequate entity knowledge, structured external information can improve generation quality. Note that this distinction is an operational proxy based on whether the encoder can retrieve the correct entity, and does not necessarily reflect whether the entity was observed during model training.

In contrast, for Hits=1, where the encoder already retrieved the correct entity (orange bars), $\Delta$DINO does not consistently remain positive. Depending on the model and the value of $k$, the improvement becomes small or slightly negative. For example, FLUX shows negative differences in some Hits=1 cases, suggesting that when the model already encodes the entity to some extent, additional descriptions do not necessarily provide further benefits and may even introduce redundancy. We observe similar patterns in SD3.5 and Qwen-Img. While unknown entities consistently benefit from augmentation, the effect on known entities depends on the model, indicating that external descriptions mainly compensate for missing internal representations rather than further strengthening already encoded knowledge.

Table 6 additionally shows that the majority of entities are categorized as unknown across all image generation models. For example, under the Hits@1 criterion, more than 99% of entities are classified as unknown for most models, indicating that current text encoders in image generation models fail to reliably retrieve the correct entity from ambiguous descriptions. Even under Hits@10, the proportion of known entities remains extremely limited. These observations further support our motivation that existing image generation models possess insufficient internal knowledge about many entities.

Despite this limitation, Figure 3 demonstrates that TEXTTIGER consistently improves DINOScore, particularly for unknown entities. In other words, even when the model fails to internally represent the target

---

[4]Among the automatic evaluation metrics, we base our analysis on DINOScore in our study, because it shows the widest score range in Tables 2 and 3, which facilitates clearer analysis.

Table 6: Distribution of known and unknown entities. An entity is categorized as known if the gold entity is included in the top-$k$ retrieval results. The percentage (%) column indicates the ratio of known entities.

| Category | Model | Hits@1 | | | Hits@5 | | | Hits@10 | | |
|---|---|---|---|---|---|---|---|---|---|---|
| | | Known | Unknown | % | Known | Unknown | % | Known | Unknown | % |
| Landmarks | Dreamlike | 1 | 2755 | 0.0 | 5 | 2751 | 0.2 | 10 | 2746 | 0.4 |
| | PixArt | 10 | 2746 | 0.4 | 25 | 2731 | 0.9 | 34 | 2722 | 1.2 |
| | FLUX | 13 | 2743 | 0.5 | 22 | 2734 | 0.8 | 34 | 2722 | 1.2 |
| | SD3.5 | 32 | 2724 | 1.2 | 57 | 2699 | 2.1 | 89 | 2667 | 3.2 |
| | Qwen-Img | 26 | 2730 | 0.9 | 65 | 2691 | 2.4 | 95 | 2661 | 3.4 |
| Paintings | Dreamlike | 1 | 2214 | 0.0 | 5 | 2210 | 0.2 | 10 | 2205 | 0.5 |
| | PixArt | 25 | 2190 | 1.1 | 43 | 2172 | 1.9 | 51 | 2164 | 2.3 |
| | FLUX | 24 | 2191 | 1.1 | 50 | 2165 | 2.3 | 64 | 2151 | 2.9 |
| | SD3.5 | 6 | 2209 | 0.3 | 26 | 2189 | 1.2 | 49 | 2166 | 2.2 |
| | Qwen-Img | 10 | 2205 | 0.5 | 20 | 2195 | 0.9 | 36 | 2179 | 1.6 |
| All | Dreamlike | 2 | 4969 | 0.0 | 10 | 4961 | 0.2 | 20 | 4951 | 0.4 |
| | PixArt | 35 | 4936 | 0.7 | 68 | 4903 | 1.4 | 85 | 4886 | 1.7 |
| | FLUX | 37 | 4934 | 0.7 | 72 | 4899 | 1.4 | 98 | 4873 | 2.0 |
| | SD3.5 | 38 | 4933 | 0.8 | 83 | 4888 | 1.7 | 138 | 4833 | 2.8 |
| | Qwen-Img | 36 | 4935 | 0.7 | 85 | 4886 | 1.7 | 131 | 4840 | 2.6 |

entity, supplementing external entity-specific descriptions effectively improves image generation quality. This result suggests that the gains achieved by TEXTTIGER do not simply come from reinforcing already memorized entities, but mainly from compensating for missing entity knowledge that is not sufficiently encoded in current text encoders.

### 6.4 Impact of Encoder Differences on Length Constraints

We analyze how differences among text encoders in image generation models affect prompt length constraints and generation performance. The image generation models used in this study employ different text encoders, and there are substantial differences in the maximum number of tokens they can process. For instance, SD3.5 support up to around 256 tokens[5], FLUX supports 512 tokens[6], and Qwen-Image can accept up to 1024 tokens[7]. At first glance, models capable of handling longer prompts may appear to be advantageous.

To verify this, we conducted experiments under controlled prompt length conditions. Specifically, for the original captions and prompts augmented with external knowledge, we summarized them so that the token length matched predefined limits, e.g., 256, 512, 1024 tokens, and prepared multiple settings aligned with the input lengths assumed by each model's text encoder. This allowed us to systematically compare the impact of prompt length on image generation performance.

As shown in Figure 4, simply increasing the prompt length leads to performance degradation across all models. This is likely because text encoders fail to properly retain important information in long inputs, resulting in information dilution and increased noise.

Furthermore, even when the encoder has a larger maximum token length, it does not necessarily make effective use of longer inputs. For example, although Qwen-Image can accept up to 1024 tokens, it shows similar performance degradation with longer prompts as observed in other models. This result suggests that the maximum processable length does not necessarily coincide with the effectively usable length.

---

[5] https://github.com/huggingface/diffusers/blob/36acdd7517733821476ff3c0b073e79ef76d8e1e/src/diffusers/pipelines/stable_diffusion_3/pipeline_stable_diffusion_3.py

[6] https://github.com/huggingface/diffusers/blob/a37f6f8394ac2a7ee8360c3abea811efe54512b1/src/diffusers/pipelines/flux/pipeline_flux.py

[7] https://github.com/huggingface/diffusers/blob/a37f6f8394ac2a7ee8360c3abea811efe54512b1/src/diffusers/pipelines/qwenimage/pipeline_qwenimage_controlnet.py

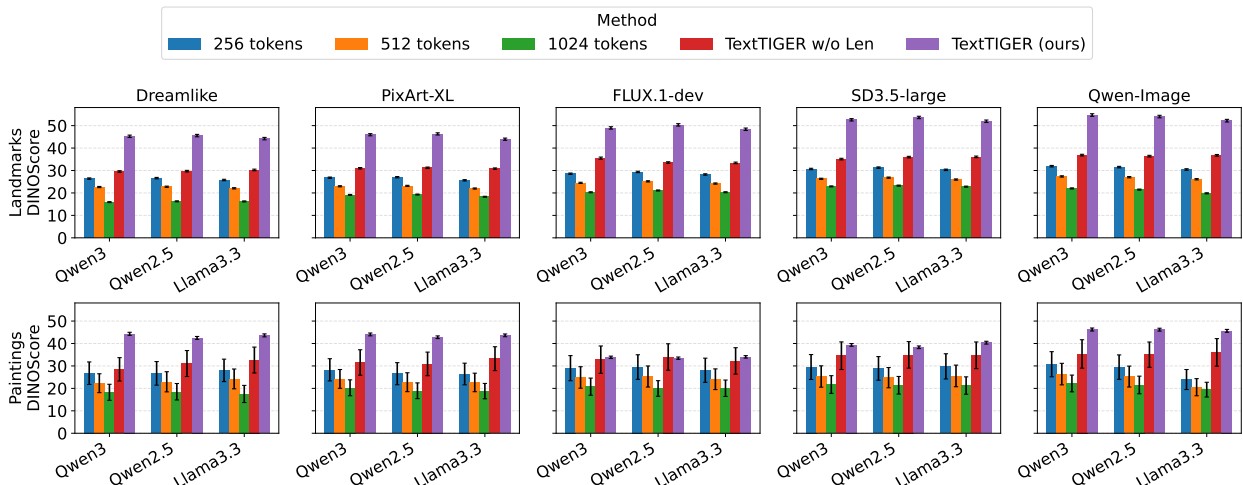

Figure 4: Ablation study. As the text encoder length differs across image generation models, we conducted a comprehensive analysis. See 6.4.

These findings indicate that in image generation models, it is important not to simply increase the input length, but to compress information according to the characteristics of the encoder.

## 6.5 Comparison of Aug-Only and RAG under a 70 Token Constraint

We compare Aug-Only, RAG, and the proposed method under conditions where the effect of prompt length is controlled. Specifically, we introduce a setting in which the generated prompts for each method are limited to a maximum of 70 tokens, defined as "RAG-70". In particular, in the RAG-70 setting, similar to TextTIGER, the generated prompts are simply truncated based on token count, ensuring that only the length is matched across methods, i.e., truncation is explicitly allowed, aiming to eliminate the influence of prompt length differences and to fairly evaluate the intrinsic performance of each method.

The results are shown in Table 7. Even when restricted to 70 tokens, Aug-Only and RAG do not show large performance improvements, and trends similar to those observed in the default setting are maintained. In other words, simply augmenting prompts with external knowledge yields only limited gains, even when the prompt length is controlled.

In contrast, TextTIGER (reports the average over five summarization models) consistently achieves high performance under the same token constraint, highlighting the importance of appropriately summarizing and compressing information, suggesting that in image generation, performance is influenced not merely by prompt length, but more critically by the quality and structure of the information.

## 6.6 Results of Human Evaluation

We randomly sample 100 instances from each of two categories, resulting in a total of 200 evaluation samples. Up to five annotators (three or more) are presented with a reference image and a pair of images generated by different methods, and they are asked to judge which image is more faithful to the reference entity using a 1–5 scale in a pairwise manner. We compare three representative methods: Cap-Only, RAG, and our proposed method TextTIGER, to analyze the effectiveness of incorporating entity knowledge and prompt refinement strategies.

Table 8 shows the results of the human evaluation. Overall, TextTIGER consistently outperforms both the Cap-Only and RAG across both categories, indicating that simply providing captions (Cap-Only) or naively augmenting external knowledge (RAG) is insufficient for accurately generating entity-aware images, whereas our approach effectively improves fidelity to the reference entities. More specifically, the Cap-Only

Table 7: DINOScore comparison between standard, 70-token, and TextTIGER settings. Each cell reports the mean with the standard deviation.

| T2I Model | RAG | | RAG-70 | | TextTIGER | |
|---|---|---|---|---|---|---|
| | **Aug-Only** | **BM25** | **Aug-Only** | **BM25** | **w/o Len (Avg.)** | **Ours (Avg.)** |
| **Landmarks** | | | | | | |
| Dreamlike | $27.05_{\pm 23.20}$ | $23.89_{\pm 22.51}$ | $30.88_{\pm 23.87}$ | $25.48_{\pm 23.38}$ | $33.55_{\pm 19.52}$ | $\mathbf{45.07}_{\pm 26.27}$ |
| PixArt | $34.84_{\pm 22.64}$ | $29.15_{\pm 23.26}$ | $35.25_{\pm 22.52}$ | $28.25_{\pm 23.15}$ | $34.96_{\pm 17.88}$ | $\mathbf{45.60}_{\pm 24.06}$ |
| FLUX | $37.44_{\pm 25.09}$ | $32.63_{\pm 25.32}$ | $36.95_{\pm 25.08}$ | $29.10_{\pm 25.20}$ | $40.63_{\pm 20.09}$ | $\mathbf{49.27}_{\pm 26.39}$ |
| SD3.5 | $35.25_{\pm 24.54}$ | $30.68_{\pm 24.59}$ | $37.34_{\pm 25.17}$ | $30.70_{\pm 25.35}$ | $40.26_{\pm 19.45}$ | $\mathbf{52.93}_{\pm 26.54}$ |
| Qwen-Img | $20.24_{\pm 22.50}$ | $18.20_{\pm 20.79}$ | $42.33_{\pm 29.00}$ | $34.29_{\pm 28.75}$ | $41.32_{\pm 19.84}$ | $\mathbf{54.02}_{\pm 27.30}$ |
| **Paintings** | | | | | | |
| Dreamlike | $27.45_{\pm 27.09}$ | $26.64_{\pm 26.09}$ | $38.36_{\pm 29.68}$ | $34.52_{\pm 24.74}$ | $34.14_{\pm 26.10}$ | $\mathbf{42.42}_{\pm 23.01}$ |
| PixArt | $39.98_{\pm 25.27}$ | $37.74_{\pm 25.58}$ | $39.02_{\pm 27.78}$ | $39.42_{\pm 28.79}$ | $35.50_{\pm 24.75}$ | $\mathbf{42.61}_{\pm 21.33}$ |
| FLUX | $32.08_{\pm 24.56}$ | $30.60_{\pm 23.99}$ | $31.10_{\pm 29.09}$ | $37.20_{\pm 24.80}$ | $36.84_{\pm 27.91}$ | $\mathbf{40.83}_{\pm 18.26}$ |
| SD3.5 | $32.52_{\pm 28.52}$ | $32.22_{\pm 27.17}$ | $31.27_{\pm 33.85}$ | $35.49_{\pm 28.78}$ | $38.61_{\pm 28.19}$ | $\mathbf{39.95}_{\pm 20.83}$ |
| Qwen-Img | $24.97_{\pm 23.51}$ | $26.04_{\pm 22.98}$ | $42.71_{\pm 29.58}$ | $40.04_{\pm 29.84}$ | $39.48_{\pm 27.68}$ | $\mathbf{45.68}_{\pm 22.46}$ |

Table 8: Pairwise human evaluations by multiple annotators. Using reference images and the corresponding generated images from each method, we rated on a 1—5 scale by up to five evaluators (three or more), along with the calculation of Kappa agreement. For summarization, Llama 3.3 was used for the Landmarks category, and Qwen 2.5 was used for the Paintings category. Dlike, SD3.5, and Q-Img represent Dreamlike, Stable Diffusion 3.5, and Qwen-Image, respectively.

| Category | Cap-Only | | | RAG | | | TextTIGER (Ours) | | |
|---|---|---|---|---|---|---|---|---|---|
| | Dlike | SD3.5 | Q-Img | Dlike | SD3.5 | Q-Img | Dlike | SD3.5 | Q-Img |
| **Evaluated Score** | | | | | | | | | |
| Landmarks | 2.28 | 1.05 | 1.55 | 1.78 | 2.17 | 2.42 | **3.63** | **4.5** | **3.86** |
| Paintings | 1.45 | 2.47 | 1.35 | 2.06 | 1.44 | 2.24 | **4.21** | **4.23** | **4.26** |
| **Kappa Agreement** | | | | | | | | | |
| Landmarks | 0.81 | 0.79 | 0.73 | 0.96 | 0.64 | 0.79 | 0.61 | 0.69 | 0.84 |
| Paintings | 0.72 | 0.84 | 0.86 | 0.73 | 0.65 | 0.69 | 0.77 | 0.68 | 0.75 |

method often fails to capture entity-specific characteristics due to limited internal knowledge in the image generation model. On the other hand, while RAG introduces additional information, its performance gains are limited. This can be attributed to the fact that directly appending retrieved descriptions may introduce noise or exceed the effective input length of the text encoder, leading to suboptimal generation quality. These observations are consistent with the trends observed in automatic evaluation metrics reported in the paper.

In contrast, TextTIGER largely improves human preference scores. By summarizing entity-specific descriptions into a concise and informative form, TextTIGER provides the model with essential knowledge while maintaining an appropriate prompt length. This results in images that better reflect the visual attributes and identity of the target entities. Annotators consistently preferred images generated by TextTIGER, suggesting that the method produces outputs that are more faithful, visually coherent, and aligned with the reference images.

### 6.7 Results of the MLLM-as-a-judge

The results of MLLM-as-a-judge shown in Tables 10 and 9 demonstrate that introducing external knowledge clearly improves the entity-related capabilities of image generation models in our study. When we look at

Table 9: MLLM-as-a-judge-based KITTEN text alignment evaluation (Txt-Img). C denotes the Cap-Only (Baseline) method. Q3, Q25, L3, G4, and G5 represent image generation results using summaries produced by Qwen3, Qwen2.5, Llama3.3, GPT-4o, and GPT-5, respectively. All values are reported as mean ± standard deviation. Statistical significance is evaluated using paired bootstrap resampling (10,000 samples) (Koehn, 2004). *, **, and *** denote $p < 0.05$, $p < 0.01$, and $p < 0.001$, compared to C.

| T2I Model | Evaluator | | | | | | | | | | | | | | | | | |
| | Gemma3 | | | | | | Qwen2.5 VL | | | | | | Phi4 | | | | | |
| | C | Q3 | Q25 | L3 | G4 | G5 | C | Q3 | Q25 | L3 | G4 | G5 | C | Q3 | Q25 | L3 | G4 | G5 |
| --- | --- | --- | --- | --- | --- | --- | --- | --- | --- | --- | --- | --- | --- | --- | --- | --- | --- | --- |
| Dreamlike | 3.38 | $4.40^{***}_{\pm0.06}$ | $4.39^{**}_{\pm0.12}$ | $4.42^{***}_{\pm0.25}$ | $4.38^{***}_{\pm0.08}$ | $4.39^{***}_{\pm0.06}$ | 2.62 | $3.63^{*}_{\pm0.20}$ | $3.64_{\pm0.11}$ | $3.64^{***}_{\pm0.21}$ | $3.65^{*}_{\pm0.23}$ | $3.64^{***}_{\pm0.05}$ | 2.59 | $3.55^{***}_{\pm0.26}$ | $3.57^{**}_{\pm0.10}$ | $3.57^{**}_{\pm0.15}$ | $3.59_{\pm0.08}$ | $3.58^{***}_{\pm0.23}$ |
| PixArt | 3.11 | $4.48^{***}_{\pm0.27}$ | $4.49^{***}_{\pm0.34}$ | $4.50^{***}_{\pm0.22}$ | $4.46^{***}_{\pm0.24}$ | $4.48^{***}_{\pm0.22}$ | 2.51 | $3.72^{***}_{\pm0.06}$ | $3.73^{*}_{\pm0.14}$ | $3.74^{*}_{\pm0.12}$ | $3.72^{*}_{\pm0.13}$ | $3.72^{***}_{\pm0.16}$ | 2.52 | $3.70^{***}_{\pm0.11}$ | $3.75^{**}_{\pm0.33}$ | $3.78^{***}_{\pm0.23}$ | $3.74^{*}_{\pm0.27}$ | $3.75^{*}_{\pm0.16}$ |
| FLUX | 3.24 | $4.38^{***}_{\pm0.24}$ | $4.40^{***}_{\pm0.26}$ | $4.39^{***}_{\pm0.28}$ | $4.35^{*}_{\pm0.06}$ | $4.36^{***}_{\pm0.13}$ | 2.37 | $3.61^{*}_{\pm0.33}$ | $3.62^{***}_{\pm0.14}$ | $3.61^{***}_{\pm0.17}$ | $3.60^{***}_{\pm0.19}$ | $3.59^{***}_{\pm0.12}$ | 2.58 | $3.63^{***}_{\pm0.13}$ | $3.67^{***}_{\pm0.32}$ | $3.65^{*}_{\pm0.12}$ | $3.63^{***}_{\pm0.20}$ | $3.64_{\pm0.06}$ |
| SD 3.5 | 3.44 | $4.44^{*}_{\pm0.24}$ | $4.46^{***}_{\pm0.18}$ | $4.45_{\pm0.16}$ | $4.43^{***}_{\pm0.21}$ | $4.42^{***}_{\pm0.31}$ | 2.57 | $3.71^{*}_{\pm0.27}$ | $3.72^{***}_{\pm0.21}$ | $3.69^{*}_{\pm0.24}$ | $3.71^{*}_{\pm0.18}$ | $3.71^{*}_{\pm0.34}$ | 2.54 | $3.66^{***}_{\pm0.13}$ | $3.69^{***}_{\pm0.10}$ | $3.69^{***}_{\pm0.31}$ | $3.67^{*}_{\pm0.24}$ | $3.66^{***}_{\pm0.10}$ |
| Qwen-Img | 3.43 | $4.47^{***}_{\pm0.21}$ | $4.33^{***}_{\pm0.21}$ | $4.30_{\pm0.15}$ | $4.28_{\pm0.33}$ | $4.28^{***}_{\pm0.30}$ | 2.53 | $3.70^{*}_{\pm0.07}$ | $3.59^{***}_{\pm0.33}$ | $3.52_{\pm0.20}$ | $3.57_{\pm0.28}$ | $3.57^{***}_{\pm0.09}$ | 2.65 | $3.56^{**}_{\pm0.21}$ | $3.25^{*}_{\pm0.31}$ | $3.24^{**}_{\pm0.11}$ | $3.24^{***}_{\pm0.27}$ | $3.23^{*}_{\pm0.14}$ |

Table 10: Results of the KITTEN entity alignment evaluation (Img-Img). See Table 9 for details.

| T2I Model | Evaluator | | | | | | | | | | | | | | | | | |
| | Gemma3 | | | | | | Qwen2.5 VL | | | | | | Phi4 | | | | | |
| | C | Q3 | Q25 | L3 | G4 | G5 | C | Q3 | Q25 | L3 | G4 | G5 | C | Q3 | Q25 | L3 | G4 | G5 |
| --- | --- | --- | --- | --- | --- | --- | --- | --- | --- | --- | --- | --- | --- | --- | --- | --- | --- | --- |
| Dreamlike | 2.44 | $3.47^{***}_{\pm0.06}$ | $3.50^{**}_{\pm0.12}$ | $3.47^{***}_{\pm0.25}$ | $3.47^{***}_{\pm0.08}$ | $3.47^{***}_{\pm0.06}$ | 1.43 | $2.47^{*}_{\pm0.20}$ | $2.48_{\pm0.11}$ | $2.45^{***}_{\pm0.21}$ | $2.43^{*}_{\pm0.23}$ | $2.45^{***}_{\pm0.05}$ | 2.03 | $3.09^{***}_{\pm0.26}$ | $3.11^{**}_{\pm0.10}$ | $3.07^{**}_{\pm0.15}$ | $3.07_{\pm0.08}$ | $3.06^{***}_{\pm0.23}$ |
| PixArt | 2.22 | $3.62^{***}_{\pm0.27}$ | $3.64^{***}_{\pm0.34}$ | $3.60^{***}_{\pm0.22}$ | $3.61^{***}_{\pm0.24}$ | $3.61^{***}_{\pm0.22}$ | 1.18 | $2.64^{***}_{\pm0.06}$ | $2.69^{*}_{\pm0.14}$ | $2.64^{*}_{\pm0.12}$ | $2.63^{***}_{\pm0.13}$ | $2.62^{***}_{\pm0.16}$ | 1.73 | $3.22^{***}_{\pm0.11}$ | $3.27^{***}_{\pm0.33}$ | $3.27^{***}_{\pm0.23}$ | $3.21^{*}_{\pm0.27}$ | $3.23^{*}_{\pm0.16}$ |
| FLUX | 2.35 | $3.54^{***}_{\pm0.24}$ | $3.58^{***}_{\pm0.26}$ | $3.53^{***}_{\pm0.28}$ | $3.55^{*}_{\pm0.06}$ | $3.56^{***}_{\pm0.13}$ | 1.28 | $2.53^{*}_{\pm0.33}$ | $2.60^{***}_{\pm0.14}$ | $2.50^{***}_{\pm0.17}$ | $2.51^{***}_{\pm0.19}$ | $2.53^{***}_{\pm0.12}$ | 2.05 | $3.15^{***}_{\pm0.13}$ | $3.23^{***}_{\pm0.32}$ | $3.17^{*}_{\pm0.12}$ | $3.14^{***}_{\pm0.20}$ | $3.13_{\pm0.06}$ |
| SD3.5 | 2.59 | $3.64^{*}_{\pm0.24}$ | $3.68^{***}_{\pm0.18}$ | $3.64_{\pm0.16}$ | $3.66^{***}_{\pm0.21}$ | $3.65^{***}_{\pm0.31}$ | 1.67 | $2.76_{\pm0.27}$ | $2.85^{***}_{\pm0.21}$ | $2.79^{*}_{\pm0.24}$ | $2.81^{*}_{\pm0.18}$ | $2.81^{*}_{\pm0.34}$ | 2.28 | $3.30^{***}_{\pm0.13}$ | $3.35^{***}_{\pm0.10}$ | $3.33^{***}_{\pm0.31}$ | $3.28^{*}_{\pm0.24}$ | $3.29^{***}_{\pm0.10}$ |
| Qwen-Img | 2.60 | $3.71^{***}_{\pm0.21}$ | $3.62^{***}_{\pm0.21}$ | $3.54_{\pm0.15}$ | $3.63_{\pm0.33}$ | $3.63^{***}_{\pm0.30}$ | 1.62 | $2.78^{**}_{\pm0.07}$ | $2.71^{***}_{\pm0.33}$ | $2.59_{\pm0.20}$ | $2.70_{\pm0.28}$ | $2.70^{***}_{\pm0.09}$ | 2.34 | $3.27^{***}_{\pm0.21}$ | $3.14^{***}_{\pm0.31}$ | $3.07_{\pm0.11}$ | $3.11^{***}_{\pm0.27}$ | $3.10^{*}_{\pm0.14}$ |

entity alignment (Img-Img), compared with Cap-Only, all settings that incorporate external knowledge through summaries generated by Qwen3, Qwen2.5, Llama3.3, GPT-4o, and GPT-5 consistently achieve higher scores across all image generation models. For example, under the Gemma3 evaluator, Dreamlike improves from 2.44 in Cap-Only to approximately 3.47. PixArt increases from 2.22 to above 3.6, and SD3.5 rises from 2.59 to the high 3.6 range. We observe similar trends with Qwen2.5 VL and Phi4 as evaluators. Models that remain in the low 1–2 range under Cap-Only consistently improve to around 2.5–3.3 after adding external knowledge, indicating that generated images reflect entity-specific characteristics more accurately when compared with reference images. The externally augmented entity descriptions contribute directly to improved visual alignment.

We observe similar improvements in text alignment (Txt-Img). With the Gemma3 evaluator, Dreamlike increases from 3.38 under Cap-Only to around 4.4, and PixArt improves from 3.11 to approximately 4.48. FLUX and SD3.5 also gain nearly one full point compared with Cap-Only. These gains indicate that prompts enriched with external knowledge strengthen the consistency between textual instructions and generated images. Under Qwen2.5 VL and Phi4 evaluators, Cap-Only remains around 2.3–2.6, whereas knowledge-augmented settings rise to approximately 3.6–3.7, showing that models reflect entity information in the prompt more faithfully after augmentation. Importantly, these improvements do not depend on a specific image generation model or evaluator. Dreamlike, PixArt, FLUX, SD3.5, and Qwen-Img all outperform Cap-Only in both entity alignment and text alignment. These consistent gains suggest that augmenting entity descriptions with external knowledge systematically compensates for internal knowledge gaps.

### 6.8 Qualitative Analysis of Generated Images

Our work further verifies that the tendency aligns with the actual visual outputs by examining the generated images shown in Tables 11 and 12. Table 11 compares images generated by SD 3.5. For the Landmarks examples "Białowieża Forest" and "Po-i-Kalyan," and the Paintings examples "Sacred conversation"and "Solly Madonna," Cap-Only produces images that clearly diverge from the reference images. For instance, in "Białowieża Forest," the model generates a generic forest scene, but it fails to reflect distinctive contextual or symbolic elements. The DINO score remains as low as 3.12. In contrast, TextTIGER with Qwen3 summarization raises the score to 85.12, and with Llama3.3 to 90.21. The generated images visually exhibit a closer match to the atmosphere and composition of the reference forest scene. A similar pattern appears in "Po-i-Kalyan." Cap-Only achieves only 2.07, whereas TextTIGER improves the score to around 80. The generated image reproduces characteristic architectural elements such as domes and minarets, and

Table 11: Actual images generated by SD 3.5 alongside the reference images and their evaluation scores. The scores are reported in the order of {DINO / Gemma3 / Qwen2.5 / Phi4}.

| Pattern | Landmarks | | Paintings | |
| --- | --- | --- | --- | --- |
| | Białowieża Forest | Po-i-Kalyan | Sacred conversation | Solly Madonna |
| Ref. Img |  |  |  |  |
| Cap-Only |  |  |  |  |
| | 3.12 / 2 / 2/ 1 | 2.07 / 1 / 1 / 1 | 3.25 / 1 / 1 / 1 | 3.18 / 2 / 1 / 1 |
| **Proposed Method (TextTIGER)** | | | | |
| Qwen3 |  |  |  |  |
| | 85.12 / 4 / 4 / 3 | 80.25 / 4 / 5 / 3 | 85.75 / 4 / 4 / 3 | 84.50 / 4 / 4 / 4 |
| Llama3.3 |  |  |  |  |
| | 90.21 / 4 / 4 / 4 | 82.23 / 4 / 5 / 3 | 78.234 / 4 / 4 / 3 | 52.12 / 1 / 2 / 2 |

this visual improvement corresponds directly to the large gains in automatic evaluation. In the Paintings category, "Sacred conversation" and "Solly Madonna" show abstract or distorted outputs under CAP-ONLY. After applying TEXTTIGER, the images clearly depict structured religious compositions with coherent figure arrangements. Both DINO and MLLM-based scores (Gemma3, Qwen2.5, Phi4) rise to around 4, matching the visible improvements. Table 12 presents additional examples using Qwen3 summarization. For "Ca' Vendramin Calergi," CAP-ONLY with FLUX yields a near failure case with a score of 1.03. After applying TEXTTIGER, the score increases to 56.23 with Dreamlike, 60.06 with SD3.5, and 89.13 with Qwen-Img. Visually, CAP-ONLY produces an incorrect portrait-like image, which indicates entity misidentification. TEXTTIGER instead generates a Venetian palace façade that matches the target entity. The increase in automatic metrics aligns with improved entity recognition. "Freedom from Want" exhibits the same pattern. CAP-ONLY scores 3.03 and fails to depict the intended scene. TEXTTIGER raises the score to the 60–75 range and successfully reconstructs the iconic family dining composition. DINO and MLLM-based evaluations, many around 4 points, match the visual improvements.

## 7 Conclusion

We addressed the limitations of current text-to-image generation models in handling entity-specific knowledge, which is essential for producing user-intended outputs. To validate this problem, we introduced a novel dataset that enriches image–caption pairs with entity annotations and detailed descriptions. Leveraging this dataset, we proposed TEXTTIGER, a method that augments prompts with external information and uses LLMs to summarize the information, ensuring the inclusion of essential knowledge while keeping the prompt within a length suitable for image generation models. Our experiments demonstrated that TEXTTIGER consistently outperforms baseline approaches across both automatic metrics MLLM-as-a-judge, along with

Table 12: Actual generated images produced using summaries by Qwen3, along with the corresponding reference images and their evaluation scores. See Table 11 for details.

| Pattern | Landmarks | | Paintings | |
|---|---|---|---|---|
| | Ca' Vendramin Calergi | Palazzo Grassi | Freedom from Want | Palazzo Dario |
| Ref. Img |  |  |  |  |
| Cap-Only (by FLUX) |  1.03 / 1 / 1/ 1 |  57.23 / 4 / 3 / 4 |  3.03 / 1 / 1 / 1 |  40.23 / 2 / 2 / 3 |
| **Proposed Method (TextTIGER)** | | | | |
| Dreamlike |  56.23 / 4 / 4 / 4 |  60.45 / 4 / 3 / 3 |  75.93 / 3 / 3 / 3 |  60.23 / 4 / 4 / 4 |
| SD3.5 |  60.06 / 3 / 4 / 4 |  75.75 / 3 / 3 / 4 |  60.23 / 4 / 4 / 4 |  60.22 / 4 / 3 / 5 |

human evaluation by multiple annotators. These results confirm that entity-aware prompt refinement is a promising direction for improving reliability.

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

## A Future Work

This study leaves several directions for future work.

First, we acknowledge the absence of human evaluation. In our work, we demonstrated that our method improves image generation performance using KITTEN, an entity-based evaluation framework shown to have a strong correlation with human judgment in evaluating whether generated images contain the intended entities. Through this framework, we verified that our approach enhances entity-level controllability in image generation.

Second, regarding the choice of image generation models, we evaluated our method on 5 open models but did not include proprietary models such as GPT-Image. [8] While we demonstrated that our approach is effective across different text encoder settings, it is in principle applicable to API-based commercial models as well. However, since such models require per-generation API costs, conducting large-scale controlled experiments would necessitate substantial additional costs. For this reason, extending the evaluation to proprietary models is also left for future work.

Finally, further investigation into summarization evaluation is possible. Our results indicate that improved summarization quality leads to better image generation performance. Although automatic summarization metrics such as ROUGE (Lin, 2004) could be employed for additional analysis, higher ROUGE scores do not necessarily correspond to prompts that are optimal for image generation. Because our objective is to improve generation performance rather than maximize textual overlap with reference summaries, we did not include ROUGE-based evaluation in this study. Exploring the relationship between conventional summarization metrics and downstream image generation quality remains an interesting direction for future research.

## B Ethical Considerations

The data used in our study were created using links that were valid as of December 2025. However, because the images and articles referenced by these links depend on Wikipedia, they may be subject to edits by other users. Thus, the reproducibility of the data cannot be fully guaranteed. The code and dataset used in our work will be made publicly available upon acceptance.

Additionally, since our dataset is constructed using Wikipedia, there is a possibility that the collected entities are biased toward specific regions or countries. On the other hand, it is inherently difficult to comprehensively cover all possible entities. Therefore, we plan to address these ethical concerns by periodically expanding and updating the dataset.

---

[8] `https://developers.openai.com/api/docs/guides/image-generation/`

## C  Appendix

### C.1  Detailed Model Settings

Table 13 shows lists of detailed model names used in our experiment.

Table 13: Detailed models' name and their citations.

| Model | Params. | HuggingFace / OpenAI API Name | Citation |
|---|---|---|---|
| **Summarization Models** | | | |
| Qwen3 | 30B | Qwen/Qwen3-30B-A3B-Instruct-2507 | Yang et al. (2025) |
| Llama 3.3 | 70B | meta-llama/Llama-3.3-70B-Instruct | Grattafiori et al. (2024) |
| Qwen2.5 | 72B | Qwen/Qwen2.5-72B-Instruct | Qwen et al. (2025) |
| GPT-4o | – | gpt-4o-mini-2024-07-18 | OpenAI et al. (2024) |
| GPT-5 | – | gpt-5-nano-2025-08-07 | Singh et al. (2025) |
| **Image Generation Models** | | | |
| Dreamlike | – | dreamlike-art/dreamlike-photoreal-2.0 | Art (2023) |
| PixArt | – | PixArt-alpha/PixArt-XL-2-1024-MS | Chen et al. (2023) |
| FLUX | 12B | black-forest-labs/FLUX.1-dev | Labs (2024) |
| SD3.5 | – | stabilityai/stable-diffusion-3.5-large | Esser et al. (2024) |
| Qwen-Img | – | Qwen/Qwen-Image | Wu et al. (2025a) |
| **MLLM-as-a-judge Models** | | | |
| Gemma 3 | 4B | google/gemma-3-4b-it | Team et al. (2025) |
| Phi 4 | 6B | microsoft/Phi-4-multimodal-instruct | Microsoft et al. (2025) |
| Qwen 2.5-VL | 7B | Qwen/Qwen2.5-VL-7B-Instruct | Bai et al. (2025) |
| **Retriever Models and Embedding Models** | | | |
| BGE | 0.1B | BAAI/bge-base-en-v1.5 | Xiao et al. (2024) |
| Contriever | 0.1B | facebook/contriever | Lei et al. (2023) |
| E5 | 0.1B | intfloat/e5-base | Wang et al. (2024) |
| RoBERTa | 0.1B | FacebookAI/roberta-base | Liu et al. (2019) |
| SimCSE | 0.1B | princeton-nlp/sup-simcse-roberta-large | Gao et al. (2021) |

**Summarization Task**  All experiments were conducted using the Transformers library (Wolf et al., 2020) with the random seed fixed at 42 for reproducibility. Qwen2.5 / 3, and Llama3.3 were quantized to 4-bit precision using the bitsandbytes library (Dettmers et al., 2022).

**Image Generation Task**  All experiments were carried out using the diffusers library (von Platen et al., 2022). The image resolution was fixed at $512 \times 512$ pixels. The guidance scale was set to 4.5, the number of inference steps to 30, and the random seed to 42.

### C.2  Difference in Retriever Performance

Table 14 presents the difference in retriever performance. Preliminary experiments compared both sparse and dense retrievers to determine the most suitable retrieval method for our RAG-based setting. As shown in Table 14, BM25 achieved the highest accuracy (31.99%), slightly outperforming dense retrievers such as BGE, Contriever, and E5.

Table 14: Comparison of retriever accuracy (%).

| Retriever | Correct (%) |
|---|---|
| **Sparse retriever** | |
| BM25 | **31.99** |
| **Dense retriever** | |
| BGE | 31.58 |
| Contriever | 31.36 |
| E5 | 31.65 |

Although the performance differences are relatively small, BM25 consistently yielded the best results among the candidates. Based on this observation, we selected BM25 as the retriever for the experiments reported in the main paper.

Table 15: Results of the MLLM-as-a-judge-based KITTEN text alignment evaluation (Txt-Img) **without abstract**. C denotes the Cap-Only (Baseline) method. Q3, Q25, L3, G4, and G5 represent image generation results using summaries produced by Qwen3, Qwen2.5, Llama3.3, GPT-4o, and GPT-5, respectively. Statistical significance markers are shown for non-baseline methods.

| T2I Model | Gemma3 | | | | | | Qwen2.5 VL | | | | | | Phi4 | | | | | |
|---|---|---|---|---|---|---|---|---|---|---|---|---|---|---|---|---|---|---|
| | C | Q3 | Q25 | L3 | G4 | G5 | C | Q3 | Q25 | L3 | G4 | G5 | C | Q3 | Q25 | L3 | G4 | G5 |
| Dreamlike | 2.41 | $4.37^{**}_{\pm0.12}$ | $4.42_{\pm0.32}$ | $4.45^{***}_{\pm0.26}$ | $4.36^{***}_{\pm0.17}$ | $4.41^{***}_{\pm0.22}$ | 2.29 | $3.66^{***}_{\pm0.25}$ | $3.61^{***}_{\pm0.18}$ | $3.67^{***}_{\pm0.09}$ | $3.62^{***}_{\pm0.10}$ | $3.66^{***}_{\pm0.24}$ | 2.72 | $3.52_{\pm0.29}$ | $3.60_{\pm0.31}$ | $3.55^{***}_{\pm0.11}$ | $3.61^{***}_{\pm0.27}$ | $3.57^{*}_{\pm0.25}$ |
| PixArt | 3.08 | $4.51^{*}_{\pm0.32}$ | $4.46^{***}_{\pm0.34}$ | $4.53_{\pm0.15}$ | $4.44^{***}_{\pm0.23}$ | $4.50_{\pm0.32}$ | 2.55 | $3.75^{***}_{\pm0.14}$ | $3.70_{\pm0.25}$ | $3.76^{**}_{\pm0.25}$ | $3.69^{**}_{\pm0.17}$ | $3.74_{\pm0.09}$ | 2.49 | $3.73^{**}_{\pm0.27}$ | $3.77_{\pm0.27}$ | $3.80^{***}_{\pm0.31}$ | $3.71^{***}_{\pm0.15}$ | $3.76^{*}_{\pm0.07}$ |
| FLUX | 3.21 | $4.41^{*}_{\pm0.31}$ | $4.37^{***}_{\pm0.28}$ | $4.42_{\pm0.25}$ | $4.33^{*}_{\pm0.32}$ | $4.38^{***}_{\pm0.27}$ | 2.40 | $3.64^{***}_{\pm0.18}$ | $3.59^{***}_{\pm0.34}$ | $3.63_{\pm0.18}$ | $3.57^{***}_{\pm0.25}$ | $3.62^{***}_{\pm0.19}$ | 2.55 | $3.66^{***}_{\pm0.13}$ | $3.64_{\pm0.26}$ | $3.68^{**}_{\pm0.32}$ | $3.60^{***}_{\pm0.33}$ | $3.66^{***}_{\pm0.32}$ |
| SD 3.5 | 3.47 | $4.41^{***}_{\pm0.27}$ | $4.49_{\pm0.10}$ | $4.47_{\pm0.10}$ | $4.40^{***}_{\pm0.07}$ | $4.45_{\pm0.22}$ | 2.54 | $3.74_{\pm0.15}$ | $3.70_{\pm0.25}$ | $3.72_{\pm0.35}$ | $3.68_{\pm0.26}$ | $3.73^{***}_{\pm0.20}$ | 2.57 | $3.69_{\pm0.32}$ | $3.66^{***}_{\pm0.06}$ | $3.72_{\pm0.33}$ | $3.64_{\pm0.20}$ | $3.68_{\pm0.10}$ |
| Qwen-Img | 3.40 | $4.44^{***}_{\pm0.29}$ | $4.36_{\pm0.20}$ | $4.33^{***}_{\pm0.16}$ | $4.25^{***}_{\pm0.33}$ | $4.30^{***}_{\pm0.14}$ | 2.56 | $3.67_{\pm0.31}$ | $3.62_{\pm0.30}$ | $3.49^{***}_{\pm0.13}$ | $3.60^{***}_{\pm0.14}$ | $3.55^{***}_{\pm0.26}$ | 2.62 | $3.59_{\pm0.20}$ | $3.28^{***}_{\pm0.12}$ | $3.27_{\pm0.19}$ | $3.20_{\pm0.20}$ | $3.26^{***}_{\pm0.30}$ |

Table 16: Results of the KITTEN entity alignment evaluation (Img-Img). See Table 15 for details.

| T2I Model | Gemma3 | | | | | | Qwen2.5 VL | | | | | | Phi4 | | | | | |
|---|---|---|---|---|---|---|---|---|---|---|---|---|---|---|---|---|---|---|
| | C | Q3 | Q25 | L3 | G4 | G5 | C | Q3 | Q25 | L3 | G4 | G5 | C | Q3 | Q25 | L3 | G4 | G5 |
| Dreamlike | 2.47 | $3.45^{***}_{\pm0.06}$ | $3.53^{***}_{\pm0.12}$ | $3.49^{***}_{\pm0.25}$ | $3.44^{***}_{\pm0.08}$ | $3.48^{**}_{\pm0.06}$ | 1.40 | $2.50^{*}_{\pm0.20}$ | $2.45_{\pm0.11}$ | $2.47^{***}_{\pm0.21}$ | $2.41_{\pm0.23}$ | $2.46^{***}_{\pm0.05}$ | 2.00 | $3.12^{***}_{\pm0.26}$ | $3.09^{**}_{\pm0.10}$ | $3.10^{**}_{\pm0.15}$ | $3.05_{\pm0.08}$ | $3.08^{***}_{\pm0.23}$ |
| PixArt | 2.25 | $3.59^{***}_{\pm0.27}$ | $3.67^{***}_{\pm0.14}$ | $3.58_{\pm0.22}$ | $3.63^{***}_{\pm0.24}$ | $3.60_{\pm0.22}$ | 1.21 | $2.61^{**}_{\pm0.06}$ | $2.72_{\pm0.14}$ | $2.66_{\pm0.12}$ | $2.60_{\pm0.13}$ | $2.65^{***}_{\pm0.16}$ | 1.70 | $3.25^{**}_{\pm0.11}$ | $3.30_{\pm0.33}$ | $3.24_{\pm0.23}$ | $3.26^{*}_{\pm0.27}$ | $3.22_{\pm0.16}$ |
| FLUX | 2.32 | $3.57^{***}_{\pm0.24}$ | $3.55^{***}_{\pm0.26}$ | $3.56^{***}_{\pm0.28}$ | $3.52^{*}_{\pm0.06}$ | $3.58_{\pm0.13}$ | 1.30 | $2.56_{\pm0.33}$ | $2.57^{***}_{\pm0.14}$ | $2.52^{***}_{\pm0.17}$ | $2.48^{***}_{\pm0.19}$ | $2.55_{\pm0.12}$ | 2.08 | $3.18^{***}_{\pm0.13}$ | $3.20_{\pm0.32}$ | $3.14^{***}_{\pm0.12}$ | $3.16^{***}_{\pm0.20}$ | $3.11_{\pm0.06}$ |
| SD3.5 | 2.56 | $3.67_{\pm0.24}$ | $3.66^{***}_{\pm0.18}$ | $3.68_{\pm0.16}$ | $3.63^{***}_{\pm0.21}$ | $3.66^{***}_{\pm0.31}$ | 1.70 | $2.73_{\pm0.27}$ | $2.88^{***}_{\pm0.21}$ | $2.77_{\pm0.24}$ | $2.84_{\pm0.18}$ | $2.80_{\pm0.34}$ | 2.31 | $3.33^{***}_{\pm0.13}$ | $3.32^{***}_{\pm0.10}$ | $3.36_{\pm0.31}$ | $3.27_{\pm0.24}$ | $3.31_{\pm0.10}$ |
| Qwen-Img | 2.63 | $3.68^{***}_{\pm0.21}$ | $3.65_{\pm0.21}$ | $3.57_{\pm0.15}$ | $3.60_{\pm0.33}$ | $3.62^{***}_{\pm0.30}$ | 1.65 | $2.75^{***}_{\pm0.07}$ | $2.74_{\pm0.33}$ | $2.61_{\pm0.20}$ | $2.68_{\pm0.28}$ | $2.72^{***}_{\pm0.09}$ | 2.37 | $3.24_{\pm0.21}$ | $3.17_{\pm0.31}$ | $3.09^{**}_{\pm0.11}$ | $3.13^{***}_{\pm0.27}$ | $3.08_{\pm0.14}$ |

## C.3 Can a RAG Approach Serve as a Substitute?

The results in Tables 2 and 3 show that a naive RAG approach (Aug-Only, BM25) cannot sufficiently substitute our method.

For example, in the Landmarks category, CLIPScore-T for Dreamlike drops from 23.978 under Cap-Only to 20.939 with Aug-Only and 20.743 with BM25. Rather than improving performance, RAG degrades it. We observe similar trends for PixArt, FLUX, and Qwen-Img, where RAG fails to consistently outperform Cap-Only. In the Paintings category, RAG occasionally produces small improvements for certain metrics. However, it does not match the consistent and substantial gains achieved by TextTIGER, suggesting that simply retrieving and appending external documents increases noise and redundancy, which prevents the model from effectively leveraging critical entity information. Image generation models face constraints in input token length and attention allocation. Thus, directly injecting unstructured retrieval outputs does not necessarily work well.

Although RAG provides a general framework for leveraging external knowledge, it does not replace our entity-focused summarization and structured knowledge injection. Challenges such as retrieval accuracy, summarization quality, and noise reduction remain unresolved, leaving further performance improvements through RAG-style approaches for future work.

## C.4 Additional MLLM-as-a-judge Analysis

In § 5.4, for evaluation using MLLM-as-a-judge, abstracts of each entity are provided as input to compensate for the lack of entity knowledge in VLMs (Mensink et al., 2023) (see Appendix D.2). While this setting improves the reliability of the evaluation, it may lead to an overestimation of the inherent capabilities of VLMs. To examine this issue, we conduct additional experiments under a setting where abstracts are not included in the input, i.e., using only the caption and the input image and letting them evaluate.

Tables 15 and 16 show the results that the differences in evaluation scores with and without abstracts are limited. They also show no significant changes in either Text Alignment or Entity Alignment. This suggests that VLMs are not effectively utilizing the provided entity descriptions. From these findings, we conclude that current VLMs have limitations in properly understanding and leveraging entity-level information, and even when explicit descriptions are provided, they do not substantially affect the evaluation outcomes. This tendency is consistent with prior studies that highlight the difficulty of entity understanding in vision-language models (Hayashi et al., 2024; Ozaki et al., 2025; Mensink et al., 2023).

## C.5 Does the Choice of Text Encoder Affect Performance?

The results in Tables 2 and  3 indicate that the choice of text encoder influences baseline performance to some extent. However, the effect of TEXTTIGER consistently appears across encoder types. First, we examine Dreamlike and SD3.5, which use CLIP as the text encoder. When we move from CAP-ONLY to TEXTTIGER, all evaluation metrics improve substantially. For instance, in Landmarks, CLIPScore-T for Dreamlike increases from 23.978 to a maximum of 25.296, and for SD3.5 from 23.882 to 25.475. We observe similar gains in CLIPScore-I, DINOScore, and PickScore, indicating that external knowledge injection strongly enhances entity understanding even for CLIP-based encoders.

Next, we analyze PixArt, which uses the T5 encoder. Under CAP-ONLY, their scores remain comparable to or slightly lower than CLIP-based models. However, applying TEXTTIGER consistently improves performance. In Landmarks, CLIPScore-T for PixArt rises from 19.106 to 23.244, and DINOScore and PickScore show similar improvements. Our result observes the same pattern in Paintings, indicating that entity-specific knowledge augmentation benefits T5-based encoders as well. Therefore, the primary driver of performance gains lies in external knowledge injection rather than in any specific encoder architecture.

## C.6 Detailed Annotation Procedure and Cost

The experiments incurred a total cost of approximately 220 US dollars through Amazon Mechanical Turk (MTurk) (Crowston, 2012). We hired workers who have an approval rate greater than 90% with at least 50 approved HITs, following the prior research (Sakai et al., 2024). Also, we intentionally included dummy questions, and if an annotator answered any of them incorrectly, we rejected their annotations unconditionally, thereby ensuring validity. Figure 5shows an example screenshot from MTurk, and Appendix D.3 provides detailed instructions.

For the retrieval task, we additionally conducted a human evaluation using Amazon Mechanical Turk (MTurk) (Crowston, 2012), with a total cost of approximately 240 US dollars. We recruited five annotators for each instance, and the final score was computed by averaging the annotators' responses. Figures 6 and  7 show the instructions and the example of questions.

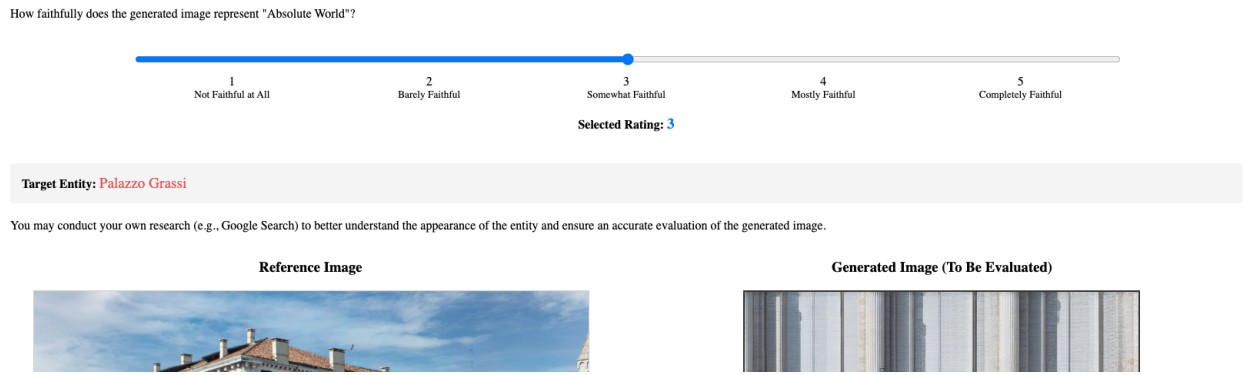

Figure 5: An example screenshot in the human evaluation from MTurk.

**Entity Selection Task**

You will see 20 short ambiguous descriptions. For each question, choose the entity, artwork, landmark, or item that best matches the description.

Each question has five answer choices. If none of the listed choices correctly matches the description, select **None of the above**.

Please read each description carefully before answering.

Figure 6: A title of the retrieval task in § 6.2 from MTurk.

**Question 1**

**Description:**

A circle on the eastern side of Paris, between the Place de la Bastille and the Bois de Vincennes, on the border of the 11th and 12th arrondissements. Widely known for having the most active guillotines during the Revolution, the square acquired its current name on Bastille Day, 14 July 1880, under the Third Republic.

⦿ Place de la Bastille
◯ Palais Rohan, Strasbourg
◯ Arc de Triomphe
◯ Palais des Papes
◯ None of the above

Figure 7: An example screenshot of the retrieval task in § 6.2 from MTurk.

## C.7   Examples of Ambiguous Abstract

Table 17 presents ambiguous prompts used in our evaluation, while Table 18 reports the corresponding Hits@1 results for each model (text encoder). IDs beginning with "L" denote samples retrieved from the Landmarks category, whereas IDs beginning with "P" denote samples retrieved from the Paintings category.

## C.8   Examples of Our Created Dataset

Tables 19 and  20 show examples of our created dataset.

## C.9   Error Analysis

Despite the overall improvements, we observe a consistent failure pattern, particularly when the target entity corresponds to a person's name. As illustrated in Table 21, cases such as "William III" and "Emile Bernard" show that models often fail to generate images that correctly reflect the identity of the person. Instead, they tend to produce generic or stylistically inconsistent portraits.

We attribute this limitation to the weak entity representations in text encoders, which struggle to capture fine-grained identity information. This issue is more obvious for paintings than for landmarks, as human-related entities require precise visual and contextual knowledge that is not sufficiently encoded.

Table 17: Selected samples.

| ID | Reference | Ambiguous Prompt |
|----|-----------|------------------|
| L1 | Palace of Versailles | A former royal residence commissioned by King Louis XIV located in Versailles, about 18 kilometres (11 mi) west of the city centre of Paris, in the Yvelines department of Île-de-France region in France. |
| L2 | Redwood National and State Parks | The parks are a complex of one United States national park and three California state parks located along the coast of northern California. The combined area contains one national park and three state parks. The parks' 139,000 acres preserve 45 percent of all remaining old-growth coast redwood forests. |
| L3 | Old Quebec | A historic neighbourhood of Quebec City, Quebec, Canada. Comprising the Upper Town and Lower Town, the area is a UNESCO World Heritage Site. Administratively, it is part of the Vieux-Québec–Cap-Blanc–colline Parlementaire district in the borough of La Cité-Limoilou. |
| P1 | Hyde Park, London | A 350-acre (140-hectare), historic Grade I-listed urban park in Westminster, Greater London. A Royal Park, it is the largest of the parks and green spaces that form a chain from Kensington Palace through Kensington Gardens and the park, via Hyde Park Corner and Green Park, past Buckingham Palace to St James's Park. It is divided by the Serpentine and the Long Water lakes. |
| P2 | Port of Hamburg | The seaport on the river Elbe in Hamburg, Germany, is 110 kilometres (68 mi) from its mouth on the North Sea. |
| P3 | Theodore Roosevelt | Jr., also known as Teddy or T. R., was the 26th president of the United States, serving from 1901 to 1909. Previously was involved in New York politics, including serving as the state's 33rd governor for two years. He served as the 25th vice president under President William McKinley for six months in 1901, assuming the presidency after McKinley's assassination. As president, emerged as a leader of the Republican Party and became a driving force for anti-trust and Progressive Era policies. |

Table 18: Top-1 retrieval results for the selected samples. IDs correspond to Table 17.

| ID | Dreamlike | PixArt | FLUX | SD3.5 | Qwen-Img | RoBERTa | SimCSE |
|---|---|---|---|---|---|---|---|
| L1 | Santorini | Par force hunting landscape in North Zealand | Disneyland | Royal Palace of La Granja de San Ildefonso | Heritage of Mercury. Almadén and Idrija | Palace of Versailles | Palace of Versailles |
| L2 | Santorini | Mausoleum of Khoja Ahmed Yasawi | Disneyland | Garajonay National Park | Rock Paintings of Sierra de San Francisco | Redwood National and State Parks | Redwood National and State Parks |
| L3 | Santorini | The Vineyard Landscape of Piedmont: Langhe-Ro... | Disneyland | Old City of Zamość | Heritage of Mercury. Almadén and Idrija | Old Quebec | Old Quebec |
| P1 | Madonna | A Converted British Family Sheltering a Chris... | Morning | The Banks of the Oise near Pontoise | Winter Landscape with Skaters | Hyde Park, London | Hyde Park, London |
| P2 | Madonna | The Banquet of the Officers of the St George ... | Morning | Seaport with the Embarkation of Saint Ursula | A View of Het Steen in the Early Morning | Port of Hamburg | Port of Hamburg |
| P3 | Madonna | Lionel Sackville, 1st Duke of Dorset | Morning | Rienzi vowing to obtain justice for the death... | M. Carey Thomas | The Third of May 1808 | William Howard Taft |

Table 19: An example of the created dataset in Paintings category.

| Entity | Description | Ref. Image |
|---|---|---|
| Queen Victoria | Victoria was Queen of the United Kingdom of Great Britain and Ireland from 20 June 1837 until her death in 1901. Her reign of 63 years and 216 days, which was longer than those of any of her predecessors, constituted the Victorian era, a period of industrial, political, scientific, and military change within the United Kingdom marked by a great expansion of the British Empire. In 1876, the British parliament voted to grant her the additional title of Empress of India. | |
| British Empire | The British Empire comprised the dominions, colonies, protectorates, mandates, and other territories ruled or administered by the United Kingdom and its predecessor states. It began with the overseas possessions and trading posts established by England in the late 16th and early 17th centuries, and colonisation attempts by Scotland during the 17th century. At its height in the 19th and early 20th centuries, it became the largest empire in history and, for a century, was the foremost global power. By 1913, the British Empire held sway over 412 million people, 23 percent of the world population at the time, and by 1920, it covered 35.5 million km2 (13.7 million sq mi), 24 per cent of the Earth's total land area. As a result, its constitutional, legal, linguistic, and cultural legacy is widespread. At the peak of its power, it was described as "the empire on which the sun never sets", as the sun was always shining on at least one of its territories. |  |
| Great Britain | Great Britain is an island in the North Atlantic Ocean off the north-west coast of continental Europe, consisting of the countries England, Scotland and Wales. With an area of 209,331 km2 (80,823 sq mi), it is the largest of the British Isles, the largest European island, and the ninth-largest island in the world. It is dominated by a maritime climate with narrow temperature differences between seasons. The island of Ireland, with an area 40 per cent that of Great Britain, is to the west – these islands, along with over 1,000 smaller surrounding islands and named substantial rocks, comprise the British Isles archipelago. | |
| United Kingdom of Great Britain and Ireland | The United Kingdom of Great Britain and Ireland was established by the Acts of Union in 1801 that united the Kingdom of Great Britain and the Kingdom of Ireland into one sovereign state. It continued in this form until 1927, when it evolved into the United Kingdom of Great Britain and Northern Ireland, after the Irish Free State gained a degree of independence in 1922. | |
| Victorian era | In the history of the United Kingdom and the British Empire, the Victorian era was the reign of Queen Victoria, from 20 June 1837 until her death on 22 January 1901, although slightly different definitions are sometimes used. The era followed the Georgian era and preceded the Edwardian era, and its later half overlaps with the first part of the Belle Époque era of continental Europe. | |

Table 20: An example of the created dataset in Landmarks category.

| Entity | Description | Ref. Image |
|--------|-------------|------------|
| Taj Mahal | The Taj Mahal is an ivory-white marble mausoleum on the right bank of the river Yamuna in Agra, Uttar Pradesh, India. It was commissioned in 1631 by the fifth Mughal emperor, Shah Jahan, to house the tomb of his beloved wife, Mumtaz Mahal; it also houses the tomb of Shah Jahan himself. The tomb is the centre-piece of a 17-hectare (42-acre) complex, which includes a mosque and a guest house, and is set in formal gardens bounded on three sides by a crenellated wall. | |
| Agra | Agra is a city on the banks of the Yamuna river in the Indian state of Uttar Pradesh, about 230 kilometres (140 mi) south-east of the national capital Delhi and 330 km west of the state capital Lucknow. It is also the part of Braj region. With a population of roughly 1.6 million, Agra is the fourth-most populous city in Uttar Pradesh and twenty-third most populous city in India. | |
| India | India, officially the Republic of India, is a country in South Asia. It is the seventh-largest country by area; the most populous country since 2023; and, since its independence in 1947, the world's most populous democracy. Bounded by the Indian Ocean on the south, the Arabian Sea on the southwest, and the Bay of Bengal on the southeast, it shares land borders with Pakistan to the west; China, Nepal, and Bhutan to the north; and Bangladesh and Myanmar to the east. In the Indian Ocean, India is near Sri Lanka and the Maldives; its Andaman and Nicobar Islands share a maritime border with Myanmar, Thailand, and Indonesia. |  |
| Marble | Marble is a metamorphic rock consisting of carbonate minerals (most commonly calcite ($CaCO_3$) or dolomite ($CaMg(CO_3)_2$) that have recrystallized under the influence of heat and pressure. It has a crystalline texture, and is typically not foliated (layered), although there are exceptions. | |
| Mausoleum | A mausoleum is an external free-standing building or standalone structure constructed as a monument enclosing the burial chamber of a deceased person or people. A mausoleum without the person's remains is called a cenotaph. A mausoleum may be considered a type of tomb, or the tomb may be considered to be within the mausoleum. | |
| Mosque | A mosque, also called a masjid, is a place of worship for Muslims. The term usually refers to a covered building, but can be any place where Islamic prayers are performed; such as an outdoor courtyard. | |
| Mumtaz Mahal | Mumtaz Mahal was the empress consort of Mughal Empire from 1628 to 1631 as the chief consort of the fifth Mughal emperor, Shah Jahan. The Taj Mahal in Agra, often cited as one of the Wonders of the World, was commissioned by her husband to act as her tomb. | |

Table 21: Examples of failure cases.

| Pattern | Landmarks | | Paintings | |
| --- | --- | --- | --- | --- |
| | Neolithic flint mines of Spiennes | Sheffield Town Hall | William III | Émile Bernard |
| Ref. Img |  |  |  |  |
| Dreamlike |  |  |  |  |
| SD3.5 |  |  |  |  |
| Qwen-Img |  |  |  |  |

# D  Prompts

We list the prompts used during experiments below.

## D.1  Prompts for Summarization Task

---

**Prompt for Summarization (TEXTTIGER)**

You must generate **ONLY ONE THING**:
a single English prompt for an image-generation model.
Do NOT output explanations, comments, apologies, thoughts, or any other text.

### HARD OUTPUT FORMAT CONSTRAINT
You MUST output **ONLY** the following block EXACTLY in this format:

<SummaryStart>
English prompt for the image generator, within 70 tokens, nothing else
<SummaryEnd>

- No additional text before or after the tags.
- No reasoning steps.
- No markdown.
- No prefaces or suffixes.
- No self-talk.
- No comments.
- No variable placeholders.

### TASK (STRICT)
Create the **optimal** English prompt for generating an iconic image of **{title}**.

Use only information logically inferable from the summary below.
Assume the image-generation model does NOT know what "{title}" is.

### REQUIREMENTS
- Length: ** 70 tokens**
- Include **all concrete visual details** required for correct generation:
- environment (sea, mountains, city, interior, etc.)
- physical structure, shapes
- materials, colors
- atmosphere, lighting
- perspective or composition
- style (only if described or inferable)
- measurements (height/width) if included in the summary
- Do NOT include:
- citations
- mentions of "summary," "tokens," or the instructions
- analysis or meta text
- any text outside <SummaryStart> ⋯ <SummaryEnd>
- Generate
- ONLY prompt
- ONLY in English
- ONLY 1 sentence

### REFERENCE SUMMARY
Below is the summary of {abstract_tokens} tokens.
Use ONLY the information contained in it.

— SUMMARY BELOW —
{abstract}
— END SUMMARY —

Now output ONLY the required block:

<SummaryStart>
...
<SummaryEnd>

<SummaryStart>

---

## Prompt for Summarization (TEXTTIGER W/O LEN)

You must generate **ONLY ONE THING**:
a single English prompt for an image-generation model.
Do NOT output explanations, comments, apologies, thoughts, or any other text.

### HARD OUTPUT FORMAT CONSTRAINT
You MUST output **ONLY** the following block EXACTLY in this format:

<SummaryStart>
English prompt for the image generator, nothing else
<SummaryEnd>

- No additional text before or after the tags.
- No reasoning steps.
- No markdown.
- No prefaces or suffixes.
- No self-talk.
- No comments.
- No variable placeholders.

### TASK (STRICT)
Create the **optimal** English prompt for generating an iconic image of **{title}**.

Use only information logically inferable from the summary below.
Assume the image-generation model does NOT know what "{title}" is.

### REQUIREMENTS
- Include **all concrete visual details** required for correct generation:
- environment (sea, mountains, city, interior, etc.)
- physical structure, shapes
- materials, colors
- atmosphere, lighting
- perspective or composition
- style (only if described or inferable)
- measurements (height/width) if included in the summary
- Do NOT include:
- citations
- mentions of "summary" or the instructions
- analysis or meta text
- any text outside <SummaryStart> ⋯ <SummaryEnd>
- Generate
- ONLY prompt
- ONLY in English

— SUMMARY BELOW —
{abstract}
— END SUMMARY —

Now output ONLY the required block:

<SummaryStart>
...
<SummaryEnd>

<SummaryStart>

## D.2   Prompts for MLLM-as-a-judge (KITTEN)

---

**Prompt for Entity Alignment**

In this task, you will be provided with a Prompt, Reference Images, and a Generated Image. Your job is to assess the factual accuracy of the generated image with respect to the prompt and the reference images. The goal is to ensure that the entity described in the prompt is factually correct and accurately represented.

First, evaluate how faithfully the generated image represents the reference entity.
Consider whether the key features and overall appearance of the reference entity are accurately depicted.
Question 1: How faithfully does the generated image represent the entity mentioned in the prompt?

Candidate Answers:
1 (Not faithful at all): The generated image does not represent the reference entity at all.
There are no discernible visual similarities to the reference entity.
2 (Barely faithful): The generated image faintly represents the reference entity, with significant effort needed to see any resemblance.
Minor visual elements may be present, but crucial features or characteristics are missing or significantly misrepresented.
3 (Somewhat faithful): The generated image somewhat represents the reference entity, but it's not prominent. There is a clear visual connection in terms of composition, style, or some key elements, but there are noticeable differences, omissions, or misinterpretations.
4 (Mostly faithful): The generated image mostly represents the reference entity and clearly presents it. The generated image draws strong visual inspiration with a strong connection in terms of overall composition, style, key elements, and/or subject matter, despite some variations in details.
5 (Completely faithful): The generated image fully represents the reference entity accurately.
It captures all key elements, composition, and style in a way that is almost identical to the reference entity.

Answer in the exact format below:
Question 1:
Answer: [1-5]
Reason: [Provide a clear explanation for your answer]

Caption: `caption`
Abstract: `abstract` (We remove this part when conducting Appendix C.4.)
Image: `image`

---

**Prompt for Prompt Alignment**

In this task, you will be provided with a Prompt and a Generated Image.

Evaluate how well the generated image captures all aspects described in the prompt.
Focus on background elements, contextual details, materials, styles, and other visual features.

Question 2: How well does the generated image depict the details described in the prompt?

Candidate Answers:
1 (Not at all): None of the described elements are present in the image.
2 (Slightly): A few minor elements are present, but most are missing or inaccurate.
3 (Moderately): Some elements are present and somewhat accurate, but others are missing or misrepresented.
4 (Mostly): Most of the described elements are clearly and accurately depicted.
5 (Completely): All relevant aspects of the prompt are thoroughly and accurately represented.

Answer in the exact format below:
Question 2:
Answer: [1-5]
Reason: [Provide a clear explanation for your answer]

Caption: `caption`
Abstract: `abstract` (We remove this part when conducting Appendix C.4.)
Image: `image`

---

## D.3   Instruction for Human Evaluation

### Instruction for the generated image evaluation

You will be shown a reference image and a generated image.
Your task is to compare them and evaluate how faithfully the generated image represents the real-world entity.
Please consider whether the shape, structure, colors, and other important visual characteristics are accurately depicted.

Rating Scale

1 (Not Faithful at All):
The generated image does not represent the reference entity in any meaningful way.
There is no noticeable visual similarity.

2 (Barely Faithful):
The generated image only faintly resembles the reference entity.
Recognizing similarities requires considerable effort.
Some minor visual elements may be present, but key features are missing or significantly misrepresented.

3 (Somewhat Faithful):
The generated image shows a moderate resemblance to the reference entity.
There is a clear visual connection in certain elements,
but noticeable differences, omissions, or inaccuracies remain.

4 (Mostly Faithful):
The generated image clearly represents the reference entity.
While there may be minor variations in details, the overall structure, composition, and major elements strongly correspond to the reference.

5 (Completely Faithful):
The generated image accurately and fully represents the reference entity.
All major elements, structure, and overall appearance closely match the reference.

You may conduct your own research (e.g., Google Search) to better understand the appearance of the entity and ensure an accurate evaluation of the generated image.

### Instruction for the retrieval task

You will see 20 short ambiguous descriptions.
For each question, choose the entity, artwork, landmark, or item that best matches the description.

Each question has five answer choices.
If none of the listed choices correctly matches the description, select **None of the above**.

Please read each description carefully before answering.

Evaluation Guidelines

- Focus on identifying the entity that most accurately matches the description.
- Consider key attributes such as appearance, function, historical background, cultural relevance, or notable characteristics mentioned in the description.
- Some descriptions may intentionally be ambiguous or partially informative, so choose the option that best fits overall.
- If no candidate adequately matches the description, choose **None of the above**.

