# OpenReview forum: "TextTIGER: Text-based Intelligent Generation with Entity Prompt Refinement for Text-to-Image Generation"
_TMLR — Rejected by TMLR_

### Review · Reviewer_K3b8 · 2026-04-22

**Summary Of Contributions:**

The authors present TextTIGER, a training-free framework that refines prompts for text-to-image models to better generate specific entities. The method retrieves Wikipedia descriptions for entities and uses an LLM to summarize them into a short text, which is appended to the original prompt. They also introduce a dataset of image-caption pairs with entity annotations.

Key strengths include a well-designed probing experiment showing that standard text encoders fail at entity retrieval, and a clear demonstration that simply appending raw retrieved text degrades performance due to attention dilution.

Key weaknesses involve limited algorithmic novelty, an overly rigid and arbitrary token-length constraint, and a problematic evaluation protocol relying entirely on MLLMs.

**Additional Comments:**

The internal knowledge probing experiment on text encoders is a very strong addition and provides excellent motivation. However, the algorithmic novelty is somewhat incremental. Addressing the evaluation shortcomings and properly exploring the token length dynamics will significantly elevate the technical depth of this contribution.

**Audience:**

Yes

**Audience Explanation:**

The finding that text encoders are highly sensitive to token length and perform poorly on raw retrieved text is practically relevant for anyone building retrieval-augmented systems for image generation. The released dataset will also benefit the community.

**Claims And Evidence:**

No

**Claims Explanation:**

The central claim that TextTIGER genuinely generates user-intended outputs with superior visual fidelity is not fully convincing in its current state.

First, the authors rely entirely on automated metrics and MLLM-based evaluations. The absence of human evaluation directly contradicts the strength of their visual fidelity claims.

Second, the strict 70-token limit imposed during the LLM summarization phase is unreasonable and poorly justified. The paper utilizes diverse text encoders, such as CLIP which has a 77-token limit, and T5 which has a 512-token limit. Enforcing a universal 70-token constraint ignores these architectural differences. The authors fail to explore how varying the token length impacts the generation quality across different encoders.

**Requested Changes:**

1. Conduct a double-blind human evaluation on a representative sample of generated images to validate the MLLM-based scores and substantiate the visual fidelity claims. This is critical for acceptance. (important)

2. Perform an ablation study on the token length constraint. The strict 70-token limit needs empirical justification. The authors may consider investigating and reporting the optimal range of token quantities for different text encoders employed in the experiments, so as to demonstrate how models like CLIP and T5 behave under different summary length conditions. (important)

3. Provide a qualitative failure case analysis. Discuss scenarios where the token limit is descriptively inadequate to capture complex spatial or architectural details.

---

> ### Author Response · Authors · 2026-05-20
> **Official Response from the Authors (1/2)**
>
> Thank you very much for your valuable comments. Your feedback was highly insightful and has been very helpful in improving our work.
>
> In response to your suggestions, we conducted additional experiments, analyzed the results, and incorporated the findings into the revised manuscript.
> We provide the details below.
>
> > ### Weakness 1: Conduct a double-blind human evaluation on a representative sample of generated images…
>
> Thank you for your valuable suggestion.
> Following your recommendation, **we conducted a double-blind human evaluation using Amazon Mechanical Turk (MTurk)**.
>
> Specifically, we randomly sampled 200 instances (100 from each category) and evaluated how closely the generated images match the reference images from the perspective of entity fidelity, following the evaluation protocol used in prior work [2].
>
> To ensure a fair comparison, we evaluated three methods: the baseline Cap-Only, the RAG-based approach, and our proposed method, TextTIGER.
>
> For the Landmarks category, we used Qwen2.5 as the summarization model, while for the Paintings category, we used Qwen3.
>
> We chose these models to reflect realistic deployment scenarios, where high-performance models are typically used, and thus provide the most meaningful evaluation setting.
>
> Annotators were presented with image pairs (reference and generated images) and asked to rate their similarity on a 1–5 scale in a pairwise evaluation setting.
>
> We hired workers who have an approval rate greater than 90% with at least 50 approved HITs, following the prior research. [3]
>
> **As a result, the inter-annotator agreement measured by Cohen’s Kappa ranges from 0.61 to 0.96 for Landmarks and from 0.65 to 0.86 for Paintings.**
>
> These values are generally interpreted as indicating substantial agreement or higher, suggesting that the evaluation is not dominated by subjective factors but is based on sufficiently reliable and consistent judgments.
>
> In terms of human evaluation scores, TextTIGER consistently outperforms both the baseline and RAG across all settings (i.e., categories and generation models).
> In particular, for the Landmarks category, we observe improvements of more than 2 points.
>
> For the Paintings category, all models achieve scores above 4 (out of 5) when using TextTIGER.
>
> **These results demonstrate that the effectiveness of our method is supported not only by automatic metrics but also from the perspective of human-perceived visual fidelity.**
>
> Therefore, in response to the concern that our claims regarding visual fidelity are not supported by human evaluation, we believe that the additional double-blind human evaluation conducted in this rebuttal provides strong and reliable evidence of the superiority of TextTIGER.
>
> These results have been incorporated into Section 6.6, with corresponding revisions made to the main text.
>
> We also provide detailed procedures and cost information in Appendix C.6, and include the annotation instructions in Appendix D.3.
>
> [1] Crowston, Kevin. "Amazon mechanical turk: A research tool for organizations and information systems scholars." Shaping the Future of ICT Research. Methods and Approaches: IFIP WG 8.2, Working Conference, Tampa, FL, USA, December 13-14, 2012. Proceedings. Berlin, Heidelberg: Springer Berlin Heidelberg, 2012.
>
> [2] Huang, Hsin-Ping, et al. "KITTEN: A Knowledge-Integrated Evaluation of Image Generation on Visual Entities." Transactions on Machine Learning Research.
>
> [3] mCSQA: Multilingual Commonsense Reasoning Dataset with Unified Creation Strategy by Language Models and Humans (Sakai et al., Findings 2024)
>
> ---
>
> > ### Weakness 2: Perform an ablation study on the token length constraint.
>
> Thank you for this important comment.
>
> **To address this point, we conducted an ablation study on token length, and report the results in Section 6.4 and Figure 4.**
>
> **Specifically, we evaluated performance across multiple token lengths (e.g., 256, 512, and 1024 tokens).**
>
> **The results show that simply increasing the token length does not lead to consistent performance improvements.**
>
> In fact, longer inputs can even degrade performance in some cases.
>
> In contrast, TextTIGER consistently achieves strong performance across all settings by using appropriately summarized prompts, suggesting that the key factor is not merely optimizing token length, but rather how the information is effectively compressed and organized.
>
> This further supports the effectiveness of TextTIGER.

---

> > ### Author Response · Authors · 2026-05-20
> > **Official Response from the Authors (2/2)**
> >
> > > ### Weakness 3. Provide a qualitative failure case analysis.
> >
> > Thank you for your good suggestion, and **through our analysis, we identified several failure cases of TextTIGER.**
> >
> > Specifically, failures tend to occur when
> > - (1) Detailed information, such as complex structures or spatial arrangements in architecture and paintings, cannot be sufficiently captured through summarization.
> > - (2) The external knowledge associated with the entity itself is ambiguous or incomplete.
> >
> > **In particular, we observe that the model still struggles to accurately handle fine-grained elements such as human faces and textual content.**
> >
> > **These findings indicate that summarization under token constraints inherently involves a trade-off in information compression, which can limit the reproduction of fine-grained visual attributes.**
> >
> > These results clarify both the strengths and limitations of our method.
> >
> > **We include this analysis in a newly added Appendix C.9 and Table 21, titled Error Analysis.**
> >
> > ---
> >
> > We would like to express our sincere appreciation once again for the thoughtful and constructive feedback.
> >
> > We hope that the new experiments and revisions introduced in this rebuttal help better demonstrate the soundness and clarity of our work.

---

### Review · Reviewer_BEK8 · 2026-04-24

**Summary Of Contributions:**

The paper studies entity-aware text-to-image generation, focusing on prompts that contain specific named entities such as landmarks and paintings. The authors argue that current text-to-image models often lack sufficient internal knowledge about long-tail or ambiguous entities, and that simply appending long external descriptions is ineffective due to text-encoder length limitations. To address this, the paper proposes a two-step prompt refinement method which retrieves entity-specific descriptions from external sources and then uses an LLM to summarize these descriptions into a shorter prompt suitable for the target image generation model. It also constructs an entity-enriched dataset based on PopVQA, covering landmarks and paintings, with captions, reference images, entity lists, and Wikipedia-derived descriptions.

**Audience:**

Yes

**Audience Explanation:**

The problem is relevant to text-to-image generation, retrieval-augmented generation, prompt optimization, multimodal evaluation, and knowledge-intensive generation. Entity fidelity is an important and under-addressed issue in text-to-image systems, especially for long-tail landmarks, artworks, culturally specific objects, and other named entities that may not be well represented in the internal knowledge of generative models. Meanwhile, the dataset construction and empirical analysis may be useful to the community, particularly because the paper evaluates multiple text encoders and image generation models and shows that entity knowledge remains weak in many current systems.

**Broader Impact Concerns:**

Since the method relies on external knowledge sources such as Wikipedia, it may also inherit factual errors, cultural bias, coverage imbalance, or outdated information from those sources. This is particularly relevant for underrepresented cultures, contested historical entities, or entities with politically sensitive descriptions.

**Claims And Evidence:**

Yes

**Claims Explanation:**

The main strength of the paper is that it addresses a concrete and practically relevant failure mode of text-to-image systems: poor generation of specific named entities. The experimental coverage is fairly broad across generation models and summarization models, and the results consistently suggest that concise entity descriptions improve generation quality compared with captions alone or naively appended long descriptions. The dataset may also be useful for future work on entity-aware image generation.

Overall, the main empirical claim that concise entity-specific prompt refinement improves entity-aware text-to-image generation is reasonably supported by the reported experiments. The paper evaluates TextTIGER on two domains, landmarks and paintings, across five text-to-image models and multiple LLM summarizers. And the results show consistent improvements over the caption-only baseline and over simple description appending in most reported automatic metrics. Moreover, the ablation between Aug-Only, TextTIGER without explicit length control, and the full TextTIGER variant also supports the authors’ argument that prompt length control is important.

**Requested Changes:**

1. The current comparison against Aug-Only and RAG is not entirely fair because these methods append long descriptions, while TextTIGER explicitly controls prompt length. The authors should include truncated Wikipedia descriptions under the same token budget. Also, extractive summaries using the first sentence or first $N$ tokens is necessary.


2. Since text-to-image outputs are stochastic, the current tables should include confidence intervals, standard deviations, or statistical significance tests. Please report the number of generated images per prompt, random seeds, decoding/sampling settings, and whether results are averaged over multiple generations.

3. The MLLM evaluator receives entity descriptions in the prompt. This may help evaluators understand the entity, but it may also bias the evaluation toward images that match the textual description rather than the true reference entity.

4. The paper defines seen/unseen entities based on whether the text encoder retrieves the correct entity in a probing task. This is a useful proxy, but it does not necessarily correspond to whether the entity was actually seen during model training. The paper should avoid overclaiming and explicitly state that this is an operational proxy for encoder-recognized versus encoder-unrecognized entities, not a true training-data seen/unseen split.

---

> ### Author Response · Authors · 2026-05-20
> **Official Response from the Authors (1/2)**
>
> We sincerely thank you for the detailed and constructive comments.
>
> The feedback has greatly contributed to improving both the clarity and quality of our work.
>
> Based on the suggestions, **we conducted additional experiments and made corresponding revisions.**
>
> All experimental results have been carefully analyzed and incorporated into the paper.
>
> We provide the detailed responses below.
>
> > ### Weakness 1: The current comparison against …
>
> Thank you for this insightful suggestion.
>
> **Following your comment, we conducted additional experiments where we truncated the Aug-Only and RAG inputs to match the same token budget as TextTIGER, i.e., 70 tokens.**
>
> The results are reported in Table 7 and analyzed in Section 6.5.
>
> Even under this controlled setting where all methods use the same token length, Aug-Only and RAG still underperform compared to TextTIGER.
>
> **This result indicates that the performance gain does not stem merely from controlling the input length, but rather from effectively summarizing entity-specific information.**
>
> In other words, the quality of summarization, rather than the amount of information, plays a critical role.
>
> We hope these findings further support the effectiveness of our approach.
>
> ---
>
> > ### Weakness 2 (1/3): Also, extractive summaries using the first sentence or first tokens is necessary.
>
> Thank you for the comment.
>
> **The primary goal of our method is not simply to reduce the amount of information, but to reorganize entity-specific knowledge into a form that is most effective for image generation.**
>
> **In this context, simple extractive summaries, i.e., such as taking the first sentence or the first tokens, may fail to preserve important attributes of entities, e.g., appearance, location, and distinctive features, and thus, we believe that it may not align well with our objective.**
>
>
> > ### Weakness 2 (2/3): Since text-to-image outputs are stochastic, …
>
> Thank you for the valuable suggestion.
>
> **To account for the variability arising from the stochastic nature of the generation process, we have added statistical evaluations to all major experimental results (Tables 2, 3, 9, and 10).**
>
> **Specifically, for each metric, we perform paired bootstrap resampling (10,000 resamples) [1] to compute the mean and standard deviation, and we further conduct statistical significance testing.**
>
> Section 5.2 introduces this information.
>
> The results show that our proposed method, TextTIGER, achieves statistically significant improvements over the baselines in many settings.
>
> This demonstrates that our findings are not dependent on specific generation outcomes, but rather reflect consistent trends even when accounting for stochastic variation.
>
> [1] Philipp Koehn. 2004. Statistical Significance Tests for Machine Translation Evaluation. In Proceedings of the 2004 Conference on Empirical Methods in Natural Language Processing, pages 388–395, Barcelona, Spain. Association for Computational Linguistics.
>
> > ### Weakness 2 (3/3): Please report the number of generated images per prompt, random seeds, decoding/sampling settings, and whether results are averaged over multiple generations.
>
> **We have already included these details in Appendix C.1 in the original paper.**
>
> **As they are intended to ensure reproducibility, we intentionally placed them in the appendix rather than the main content.**
>
> We hope this clarifies your concern, and we kindly invite you to refer to Appendix C.1 for further details.

---

> > ### Author Response · Authors · 2026-05-20
> > **Official Response from the Authors (2/2)**
> >
> > > ### Weakness 3: The MLLM evaluator receives entity …
> >
> > Thank you for the valuable comment.
> >
> > As a background, prior studies have pointed out that vision-language models (VLMs) often lack sufficient knowledge about entities and may fail to understand them appropriately [1,2,3].
> >
> > Motivated by this, in our evaluation setup, we provide a concise entity description (abstract) to improve the stability of the evaluation.
> >
> > At the same time, as you pointed out, this design choice may introduce potential bias.
> >
> > **To address this concern, we conducted additional experiments without providing the abstract.**
> >
> > The results are reported in Tables 15 and 16, with detailed analysis in Appendix C.4.
> >
> > **We observe that without the abstract, the evaluation scores exhibit higher variance and the differences between methods become less distinguishable.**
> >
> > In contrast, when the abstract is provided, the performance gaps between methods are more consistently observed.
> > Importantly, the overall performance differences between the settings with and without the abstract are limited, and the main conclusions of our study remain unchanged.
> >
> > Therefore, we include this additional experiment as a supplementary analysis in the appendix.
> >
> > We also provide the exact prompts used in the evaluation in Appendix D.2.
> >
> > [1] Kazuki Hayashi, Yusuke Sakai, Hidetaka Kamigaito, Katsuhiko Hayashi, and Taro Watanabe. 2024. Towards Artwork Explanation in Large-scale Vision Language Models. In Proceedings of the 62nd Annual Meeting of the Association for Computational Linguistics (Volume 2: Short Papers), pages 705–729, Bangkok, Thailand. Association for Computational Linguistics.
> >
> > [2] Shintaro Ozaki, Kazuki Hayashi, Yusuke Sakai, Hidetaka Kamigaito, Katsuhiko Hayashi, and Taro Watanabe. 2025. Towards Cross-Lingual Explanation of Artwork in Large-scale Vision Language Models. In Findings of the Association for Computational Linguistics: NAACL 2025, pages 3773–3809, Albuquerque, New Mexico. Association for Computational Linguistics.
> >
> > [3] Mensink, Thomas, et al. "Encyclopedic vqa: Visual questions about detailed properties of fine-grained categories." Proceedings of the IEEE/CVF International Conference on Computer Vision. 2023.
> >
> > ---
> >
> > > ### Weakness 4: The paper defines seen/unseen entities based on whether the text encoder retrieves the correct entity in a probing task.
> >
> > Thank you for the insightful comment.
> >
> > We acknowledge that the terms “seen/unseen” used in our original submission may be misleading, as they do not refer to whether the entity was actually observed during training, but rather to an operational distinction based on whether the text encoder can identify the entity.
> >
> > **To clarify this point, we have revised the terminology to “known/unknown” throughout the paper.**
> > **In addition, we have added the following explanation in Section 6.3 to explicitly state that this distinction is a proxy derived from the probing task:**
> > ```
> > Note that this distinction is an operational proxy based on whether the encoder can retrieve the correct entity, and does not necessarily reflect whether the entity was observed during model training.
> > We believe that this revision makes the assumptions and interpretation of our analysis clearer.
> > ```
> >
> > > ### Broader Impact Concerns
> >
> > **We have already discussed this limitation in Appendix B in the original manuscript.**
> >
> > **We also noted the potential difficulty in ensuring reproducibility due to temporal changes in the data, and stated that we will release the dataset used in this study upon acceptance.**
> >
> > However, we acknowledge that the discussion did not explicitly address the issue of cultural bias.
> >
> > **To address this, we have added the following clarification to Appendix B.**
> >
> > We sincerely appreciate this important and insightful suggestion.
> >
> > ```
> > Additionally, since our dataset is constructed using Wikipedia, there is a possibility that the collected entities are biased toward specific regions or countries.
> > On the other hand, it is inherently difficult to comprehensively cover all possible entities.
> > Therefore, we plan to address these ethical concerns by periodically expanding and updating the dataset.
> > ```
> >
> > ---
> > We would like to once again express our sincere gratitude for the valuable and constructive feedback.
> >
> > We hope that the additional experiments and revisions provided in this rebuttal help improve the clarity and validity of our work.

---

### Review · Reviewer_9eNX · 2026-05-09

**Summary Of Contributions:**

This paper proposes TextTIGER for text-to-image generation with named entities. The core idea is that text-to-image models often lack sufficient knowledge about long-tail or ambiguous entities, so the method augments the original prompt with external entity descriptions, then uses an LLM to summarize those descriptions into a shorter prompt suitable for the text encoder’s length constraints. The authors also construct an entity-aware dataset by extending PopVQA with Wikipedia-derived entity descriptions for landmarks and paintings, yielding 5,009 instances. Experiments across five open text-to-image models and several summarization LLMs show consistent improvements over the baselines on different evaluation metrics. The paper further analyzes text encoder entity knowledge via retrieval-style probing and argues that external descriptions are especially helpful for entities not well represented by the model’s text encoder.

**Audience:**

Yes

**Audience Explanation:**

TextTIGER approach is simple and effective, but the evaluation approach lacks of human study to validate its effectiveness, and the scalability is limited to high-quality external knowledge. I hope the authors can add human studies to better demonstrate the effectiveness of the approach.

**Claims And Evidence:**

No

**Claims Explanation:**

**Strengths**
1. The paper addresses a real limitation of text-to-image models about how to generate named entities, such as landmarks.
2. The experimental results demonstrate the effectiveness of TextTIGER, though it is simple and intuitive.
3. It is good to see that the benefits generalize to unseen entities.


**Weaknesses**
1. The scalability and effectiveness of TextTIGER is limited to the amount of available and high-quality annotations of named entities.
2. For Section 6.2, when the retrieval performance is very low (e.g., Hits@K is low), what are the retrieved descriptions and corresponding named entities? What is a reasonable baseline (it would be good to have a small human study to validate the setup)?
3. For Section 6.3, what is a “seen” entity (do you mean Hits@100 > 0 or etc)? Also, what is the distribution between seen and unseen categories?
4. For Section 6.3, given that the models mainly gain benefits on unseen entities and sometimes negative impact on seen entities, why not adding another stage to dynamically identify whether an entity is “known or unknown” and if “unknown”, then augmented the model with TextTIGER?
5. Lack of human evaluation for the generated images to better understand the effectiveness of TextTIGER.

**Requested Changes:**

See weaknesses.

---

> ### Author Response · Authors · 2026-05-20
> **Official Response from the Authors (1/3)**
>
> We sincerely thank you for the insightful and constructive comments.
>
> We carefully took into account each suggestion and have summarized our responses, additional experiments, including human evaluations, below.
>
> We hope these revisions and clarifications address your concerns.
>
> > ### Weakness 1:  The scalability and effectiveness of TextTIGER is limited to….
>
> Thank you for pointing it out, and we agree that the effectiveness of TextTIGER depends on the availability and quality of externally retrieved entity descriptions.
>
> However, at the same time, our framework does not require manually curated annotations specific to TextTIGER.
>
> **In this work, entity descriptions are automatically obtained from publicly available knowledge sources such as Wikipedia, using entity-linked metadata already included in the dataset.**
>
> **Thus, the scalability of our method mainly depends on the availability of external knowledge sources rather than expensive human annotation.**
>
> **Moreover, recent retrieval systems and LLMs can dynamically access web scale knowledge bases, making it feasible to obtain descriptions for a large number of entities, including long-tail entities.**
>
> We believe that improving retrieval robustness and handling noisy or incomplete descriptions are important directions for future work.
>
> ---
>
> > ### Weakness 2: For Section 6.2, when the retrieval performance is very low ….
>
> Thank you for your valuable suggestion.
>
> From the initial submission, we’ve already included sentence embedding models, i.e., SimCSE and RoBERTa, in Figure 2 as (reasonable) baselines for the knowledge probing task.
>
> **However, following your suggestion, we conducted a human evaluation to further validate the retrieval setup itself to clarify your concern.**
> **Specifically, we randomly sampled 100 instances from each category (landmarks and paintings), resulting in 200 instances in total.**
>
> For the evaluation, we used SimCSE, which achieved the best retrieval performance in Figure 2, and constructed a multiple choice setting consisting of the top 4 retrieved entities together with an additional “None of the above” option.
>
> Five annotators were recruited through Amazon Mechanical Turk (MTurk) [1], and they evaluated each instance.
>
> The final score was computed by averaging their responses.
>
> **Table 5 (newly created) shows the result that the human performance achieved the highest retrieval accuracy among all baselines, suggesting that humans can reliably identify the target entity from the ambiguous abstracts.**
>
> **This indicates that the ambiguous abstracts used in our setup are not difficult but instead constitute a challenging retrieval task.**
>
> Based on these findings, we created Table 5 to include the Human baseline and additionally added detailed analysis in Section 6.2.
>
> We believe that these additional results demonstrate that the low retrieval accuracy observed for image generation model text encoders stems from limited entity understanding capability.
>
> [1] Crowston, Kevin. "Amazon mechanical turk: A research tool for organizations and information systems scholars." Shaping the Future of ICT Research. Methods and Approaches: IFIP WG 8.2, Working Conference, Tampa, FL, USA, December 13-14, 2012. Proceedings. Berlin, Heidelberg: Springer Berlin Heidelberg, 2012.

---

> > ### Author Response · Authors · 2026-05-20
> > **Official Response from the Authors (2/3)**
> >
> > > ### Weakness 3: For Section 6.3, what is a “seen” entity (do you mean Hits@100 > 0 or etc)?
> >
> > Thank you for your valuable comment.
> >
> > **First, since the definition of “seen” entities was unclear in the original manuscript, we added a more detailed explanation in Section 6.3.**
> >
> > **In this work, an entity is categorized as a “known” entity when the gold entity is included within the top-$k$ retrieval results in Hits@$k$, and otherwise categorized as an “unknown” entity as discussed in Section 6.2.**
> >
> > In addition, following comments from another reviewer that the terminology “seen/unseen” was confusing, we revised the terminology throughout the paper to “known/unknown.”
> >
> > Furthermore, following your suggestion, we added Table 6 to analyze the distribution of known and unknown entities.
> >
> > The results in Table 6 show that, across almost all image generation models, the vast majority of entities are categorized as unknown.
> >
> > For example, under the Hits@1 criterion, more than 99% of entities are classified as unknown for most models, indicating that current text encoders in image generation models fail to retrieve the correct entity from ambiguous descriptions.
> >
> > Even under Hits@10, the proportion of known entities remains extremely limited.
> >
> > These findings further support our motivation that existing image generation models do not possess sufficient internal knowledge for many entities.
> >
> > In addition, the analysis in Figure 3 shows that TextTIGER consistently improves DINOScore particularly for unknown entities.
> >
> > In other words, even when the target entity is not represented within the model, supplementing external entity specific descriptions can improve image generation quality.
> >
> > **These findings suggest that the improvements achieved by TextTIGER do not simply come from reinforcing already known entities, but rather from compensating for entity knowledge that is not encoded in current text encoders.**
> >
> > > ### Weakness 4: For Section 6.3, given that the models mainly gain benefits on unseen entities …
> >
> > Thank you for your valuable suggestion.
> >
> > We also believe that dynamically determining whether an entity is sufficiently represented within the model and selectively applying TextTIGER only to “unknown” entities is a very promising direction.
> >
> > In this work, our primary goal was to investigate whether augmenting entity specific descriptions can generally improve entity aware image generation.
> >
> > **Thus, to ensure fair comparisons across models and settings, we uniformly applied TextTIGER to all entities.**
> >
> > As shown in Section 6.3, we observed substantial improvements for unknown entities, while the improvements for known entities were more limited and occasionally slightly negative.
> >
> > These findings suggest that introducing adaptive augmentation could further improve efficiency and robustness.
> >
> > However, automatically determining whether a text encoder already possesses sufficient knowledge about a target entity is itself a challenging problem.
> >
> > One possible approach would be to use a similarity based criterion and assume that entities whose retrieval similarity exceeds a predefined threshold are already “known” to the model.
> >
> > However, such threshold based approaches would likely be highly empirical and sensitive to model specific calibration and retrieval settings.
> >
> > Another possible direction would be to introduce an additional classification head trained to predict whether an entity is known or unknown to the encoder.
> >
> > For example, if the classifier predicts that the encoder lacks sufficient entity knowledge, TextTIGER could be selectively applied only in such cases.
> >
> > However, these approaches would require additional supervision, architectural modifications, and dedicated training procedures, which are beyond the scope of the current work.
> >
> > Therefore, we believe that a dynamic framework which first estimates whether an entity is sufficiently encoded within the model and then selectively applies external knowledge augmentation based on that estimation is an important direction for future work.

---

> ### Author Response · Authors · 2026-05-20
> **Official Response from the Authors (3/3)**
>
> > ### Weakness 5: Lack of human evaluation …
>
> Thank you for your valuable comment.
>
> We received similar concerns from Reviewer K3b8 regarding the lack of human evaluation for generated images.
>
> **Following your suggestions, we conducted a double blind human evaluation using Amazon Mechanical Turk (MTurk).**
>
> Specifically, we randomly sampled 100 instances from each category, resulting in 200 instances in total, and evaluated entity fidelity between generated images and reference images.
>
> We compared Cap-Only, RAG, and our proposed method, TextTIGER.
>
> Annotators were presented with image pairs and asked to rate their similarity on a scale from 1 to 5.
>
> **As a result, TextTIGER consistently outperformed both the baseline and RAG across all settings.**
>
> **In addition, the inter-annotator agreement measured by Cohen’s Kappa was sufficiently high, indicating that the evaluation was not heavily dominated by subjective factors.**
> **These findings demonstrate that the effectiveness of TextTIGER is supported not only by automatic metrics but also from the perspective of human perceived visual fidelity.**
>
> We incorporated these results into Section 6.6 and additionally provide detailed evaluation procedures in Appendix C.6 and annotation instructions in Appendix D.3.
>
> ---
>
> We would like to sincerely thank you once again for their insightful and constructive feedback.
>
> We hope that the additional experiments and revisions included in this rebuttal further clarify and strengthen our work.

---

### Author Response · Authors · 2026-05-20
**Authors’ Response to the Reviewers**

Dear all,

Thank you very much for your valuable comments and suggestions on our paper. We sincerely appreciate the time and effort you devoted to reviewing our work.

We have updated the PDF by addressing the official comments from OpenReview. The revisions are color-coded as follows:

- Red: Suggested by Reviewer K3b8
- Blue: Suggested by Reviewer BEK8
- Green: Suggested by Reviewer 9eNX

Please kindly check the updated PDF.

Best regards,

The Authors

---

> ### Author Response · Authors · 2026-05-22
> **Brief Summary from the Authors**
>
> We would like to express our sincere gratitude to all reviewers for their insightful comments and constructive suggestions. All of your feedback greatly helped improve both the clarity and the quality of our manuscript.
>
> Today is the deadline of the discussion period, so we would like to share a summary.
>
> - In reply to Reviewer 9eNX’s suggestions, we revised the manuscript and added additional analyses and human evaluations.
> - In reply to Reviewer BEK8’s suggestions, we conducted additional controlled experiments, statistical analyses, and evaluation studies.
> - In reply to Reviewer K3b8’s suggestions, we added human evaluations, token-length ablation studies, and qualitative error analyses.
>
> In conclusion, we have made the following revisions to the manuscript.
>
> ## Revision and Expansion
>
> - Added a human evaluation for the retrieval setup using MTurk annotators and introduced a new Human baseline in Table 5. (Reviewer 9eNX)
> - Added detailed explanations of the “known/unknown” entity definition in Section 6.3 and revised the term from “seen/unseen” to “known/unknown” throughout the paper. (Reviewer 9eNX, Reviewer BEK8)
> - Added Table 6, analyzing the distribution of known and unknown entities. (Reviewer 9eNX)
> - Expanded the discussion on adaptive augmentation for dynamically determining whether entities are sufficiently represented in text encoders. (Reviewer 9eNX)
> - Conducted a double-blind human evaluation of generated images using MTurk and added the results to Section 6.6. (Reviewer 9eNX, Reviewer K3b8)
> - Added detailed human evaluation procedures in Appendix C.6 and annotation instructions in Appendix D.3. (Reviewer 9eNX, Reviewer K3b8)
> - Conducted additional experiments using truncated Aug-Only and RAG inputs under the same token budget as TextTIGER, and added the results in Table 7 and Section 6.5. (Reviewer BEK8)
> - Added statistical evaluations, including paired bootstrap resampling and significance testing, to the major experimental results in Tables 2, 3, 9, and 10. (Reviewer BEK8)
> - Added supplementary evaluation experiments without providing entity abstracts to the evaluator, and included the results in Tables 15 and 16 and Appendix C.4. (Reviewer BEK8)
> - Clarified that the “known/unknown” distinction is an operational proxy derived from the probing task rather than a true training-data split. (Reviewer BEK8)
> - Expanded the discussion of cultural bias and dataset limitations in Appendix B. (Reviewer BEK8)
> - Conducted a token-length ablation study across multiple token lengths and added the results in Section 6.4 and Figure 4. (Reviewer K3b8)
> - Added a qualitative error analysis discussing failure cases related to summarization limitations and ambiguous external knowledge, and included the analysis in Appendix C.9 and Table 21. (Reviewer K3b8)
>
> ## Clarification
> - Clarified that implementation details such as random seeds, sampling settings, and the number of generated images were already provided in Appendix C.1 for reproducibility purposes. (Reviewer BEK8)
> - Added further discussion regarding the scalability of TextTIGER and its dependence on external knowledge sources rather than manually curated annotations. (Reviewer 9eNX)
>
> We would like to once again sincerely thank all reviewers for their thoughtful and constructive feedback.
> We hope that the additional experiments, analyses, and revisions introduced in this rebuttal help further clarify and strengthen our work.

---

### Decision · Action_Editor_dakM · 2026-06-23

**Recommendation:** Reject

**Additional Comments:**

After the rebuttal, one reviewer votes for the rejection of this paper. The authors fail to convince the reviewers about his/her second comment. In particular, the reviewer expects a ablation study on the different token quantity ranges of various encoders as supplementary experiments. However, the authors only conduct a simple test on the maximum dimension number. The reviewer believes the claims made in the submission are not supported by clear evidence due to the lack of this ablation study. Therefore, Reviewer K3b8 leans rejection.

As for the other two reviewers, they lean accept. However, they do not champion the paper. In particular, they think the method is relatively simple, and some baselines, such as extractive summaries, remain only partially addressed.

Given the reviewers' comment, AE would recommend rejection.

**Audience:**

Yes

**Audience Explanation:**

The paper topic is interesting to the TMLR community.

**Claims And Evidence:**

No

**Claims Explanation:**

The strict 70-token limit imposed during the LLM-based summarization stage is both insufficiently justified and arguably inappropriate. The framework employs a variety of text encoders with substantially different context capacities, including CLIP, which supports up to 77 tokens, and T5, which allows sequences of up to 512 tokens. Applying a uniform 70-token constraint disregards these architectural differences and may unnecessarily restrict the expressive power of certain encoders. Moreover, the paper does not investigate how the choice of token length affects generation quality across different text encoders, leaving the rationale behind this design decision unclear.